# When right ventricular pressure meets volume: The impact of arrival time of reflected waves on right ventricle load in pulmonary arterial hypertension

Masafumi Fukumitsu[1,2], Joanne A. Groeneveldt[1], Natalia J. Braams[1], Ahmed A. Bayoumy[1,3], J. Tim Marcus[4] 🆔, Lilian J. Meijboom[4], Frances S. de Man[1], Harm-Jan Bogaard[1], Anton Vonk Noordegraaf[1] 🆔 and Berend E. Westerhof[1] 🆔

[1]*Department of Pulmonary Medicine, Amsterdam Universitair Medische Centra, Vrije Universiteit Amsterdam, Amsterdam Cardiovascular Sciences, Amsterdam, The Netherlands*
[2]*Department of Cardiovascular Dynamics, National Cerebral and Cardiovascular Center, Japan*
[3]*Department of Internal Medicine, Chest Unit, Suez Canal University Hospitals, Suez Canal University, Ismailia, Egypt*
[4]*Department of Radiology and Nuclear Medicine, Amsterdam Universitair Medische Centra, Vrije Universiteit Amsterdam, Amsterdam Cardiovascular Sciences, Amsterdam, The Netherlands*

Edited by: Bjorn Knollmann & Larissa Shimoda

Linked articles: This article is highlighted in a Journal Club article by Karvasarski et al. To read this article, visit https://doi.org/10.1113/JP283241.

The peer review history is available in the Supporting information section of this article (https://doi.org/10.1113/JP282422#support-information-section).

**Masafumi Fukumitsu** is a cardiologist-scientist, and obtained his PhD in cardiovascular physiology for work on pulmonary vascular impedance as right ventricular afterload in animal models of pulmonary hypertension at Osaka University Graduate School of Medicine, Japan. Based on his PhD projects, the present study was performed during his stay at Amsterdam University Medical Centre, Vrije Universiteit Amsterdam, The Netherlands, under the supervision of Dr Berend E. Westerhof and Professor Anton Vonk Noordegraaf. This project was supported by European Respiratory Society Research Fellowship.

**Abstract** Right ventricular (RV) wall tension in pulmonary arterial hypertension (PAH) is determined not only by pressure, but also by RV volume. A larger volume at a given pressure generates more wall tension. Return of reflected waves early after the onset of contraction, when RV volume is larger, may augment RV load. We aimed to elucidate: (1) the distribution of arrival times of peak reflected waves in treatment-naïve PAH patients; (2) the relationship between time of arrival of reflected waves and RV morphology; and (3) the effect of PAH treatment on the arrival time of reflected waves. Wave separation analysis was conducted in 68 treatment-naïve PAH patients. In the treatment-naïve condition, 54% of patients had mid-systolic return of reflected waves (defined as 34–66% of systole). Despite similar pulmonary vascular resistance (PVR), patients with mid-systolic return had more pronounced RV hypertrophy compared to those with late-systolic or diastolic return (RV mass/body surface area; mid-systolic return $54.6 \pm 12.6$ g m$^{-2}$, late-systolic return $44.4 \pm 10.1$ g m$^{-2}$, diastolic return $42.8 \pm 13.1$ g m$^{-2}$). Out of 68 patients, 43 patients were further examined after initial treatment. At follow-up, the stiffness of the proximal arteries, given as characteristic impedance, decreased from 0.12 to 0.08 mmHg s mL$^{-1}$. Wave speed was attenuated from 13.3 to 9.1 m s$^{-1}$, and the return of reflected waves was delayed from 64% to 71% of systole. In conclusion, reflected waves arrive at variable times in PAH. Early return of reflected waves was associated with more RV hypertrophy. PAH treatment not only decreased PVR, but also delayed the timing of reflected waves.

(Received 26 October 2021; accepted after revision 11 April 2022; first published online 14 April 2022)

**Corresponding author** B. E. Westerhof: Department of Pulmonary Medicine, Amsterdam Universitair Medische Centra, Amsterdam Cardiovascular Sciences, De Boelelaan 1117, 1081HV Amsterdam, The Netherlands. Email: b.e.westerhof@amsterdamumc.nl

**Abstract figure legend** Return of reflected waves augments right ventricular pressure in early systole, when the ventricular volume is relatively large. This increases ventricular wall tension, resulting in hypertrophy.

### Key points

- Right ventricular (RV) wall tension in pulmonary arterial hypertension (PAH) is determined not only by pressure, but also by RV volume. Larger volume at a given pressure causes larger RV wall tension.
- Early return of reflected waves adds RV pressure in early systole, when RV volume is relatively large. Thus, early return of reflected waves may increase RV wall tension. Wave reflection can provide a description of RV load.
- In PAH, reflected waves arrive back at variable times. In over half of PAH patients, the RV is exposed to mid-systolic return of reflected waves. Mid-systolic return of reflected waves is related to RV hypertrophy.
- PAH treatment acts favourably on the RV not only by reducing resistance, but also by delaying the return of reflected waves.
- Arrival timing of reflected waves is an important parameter for understanding the relationship between RV load and its function in PAH.

## Introduction

Right ventricular (RV) dysfunction is the major determinant of clinical outcome in pulmonary arterial hypertension (PAH) (van de Veerdonk et al., 2011). An accurate description of the increased arterial load is required in the investigation of RV failure in PAH. Although resistance and arterial compliance are commonly used to represent RV load, wave reflection can provide an additional description of ventricular load, allowing assessment of mechanical stress (wall stress or tension) as a function of time (Fukumitsu et al., 2020). Recently, the aim of reducing RV wall stress was proposed because it is the main driver of changes to the ventricle, wereas the RV function determines the prognosis of patients with pulmonary hypertension (Westerhof et al., 2017)

The cardiac pump generates pressure and flow waves (Westerhof et al., 2018). The pressure and flow waves travelling to the periphery are partly reflected in the remodelled lung vasculature and return to the heart. Reflected pressure adds to forward pressure, whereas reflected flow subtracts from forward flow (Nichols et al., 2011). In our prior study, arrival timing of reflected waves was proposed as a relevant component of RV load (Fukumitsu et al., 2020). RV mechanical stress is time-varying and depends on RV pressure and size. Early in systole, when the RV is still partially filled and wall tension is greatest, arrival of the reflected pressure wave has more impact (Westerhof et al., 2018). By contrast, late-systolic return of reflected waves may not contribute much to ventricular stress because ventricular size and wall tension are small (Chirinos et al., 2009). Furthermore, when reflected waves return during diastole, these waves may not add to RV pressure markedly because the pulmonary valve is closed, and RV pressure decreases steeply after pulmonary valve closure (Sagawa et al., 1988). In other words, reflected waves only have an impact when they return during early- or mid-systole, but not during late-systole or diastole. This concept was validated in chronic thromboembolic pulmonary hypertension (CTEPH), particularly when vascular obstructions are located in the proximal vasculature (Fukumitsu et al., 2020). Much less is known about the distribution of arrival times of reflected waves in PAH. It is also unclear whether the timing of arrival of reflected waves is altered by PAH treatment.

Thus, the present study aimed: (1) to investigate the prevalence of early- or mid-systolic return of reflected waves in PAH patients; (2) to elucidate the relation between reflected wave arrival time with RV morphology in treatment-naïve PAH patients; and (3) to assess the effect of PAH treatment on the arrival time of reflected waves.

## Methods

### Ethical approval

The present study was conducted retrospectively on a registry of patients with diagnosed PAH who routinely underwent right-sided heart catheterization (RHC), cardiac magnetic resonance imaging (MRI), a 6 min walk test and blood sampling. This study conformed to the standards set by the *Declaration of Helsinki*, except for registration in a database. Informed consent was given by all patients included after 2010 to use their data for scientific purposes. For patients included before 2010, the medical ethical committee waived the requirement of consenting the patients in retrospect and allowed using the clinical data collected in that period for scientific purposes (approval no. 2017.025).

### Study protocol

We screened 324 patients who were diagnosed as PAH from February 1996 until December 2018 at Amsterdam University Medical Centre, Vrije Universiteit Amsterdam (Amsterdam, The Netherlands). Wave separation analysis requires the complete digital data of both pulmonary artery (PA) pressure and flow waveforms. Thus, enrolment criteria were: (1) to have digital data of both PA and RV pressure curves at the initial diagnosis before treatment and (2) to have PA flow curves obtained by cardiac MRI in treatment-naïve condition within 28 days before or after diagnostic RHC [a median interval between RHC and MRI was 1 day: interquartile range (IQR) 0–6 days]. In the present study, 68 treatment-naïve patients were enrolled and separated into three groups according to the timing of peak reflected waves returning to the RV: a group with reflected waves during mid-systole (referred to as mid-systolic returners), a group with reflected waves during late-systole (late-systolic returners) and a group with reflected waves during diastole (diastolic returners). Early-systole was defined by 0–33% of systole, mid-systole was 34–66% of systole and late-systole was 67–100% of systole. In our study, no patients had reflected waves during early systole.

Out of 68 treatment-naïve patients, 43 patients who had digital data of PA flow and pressure waves at both baseline (under treatment-naïve condition) and 1 year of follow-up (under treatment condition) were examined to assess the effect of PAH treatment on timing of reflected waves (Fig. 1).

### Right heart catheterization

RHC was performed as described previously (Fukumitsu et al., 2020; Trip et al., 2015; van de Veerdonk et al., 2011). Briefly, under local anaesthesia, a 7-Fr balloon-tipped Swan-Ganz catheter (131HF7; Baxter Healthcare Corp., Irvine, CA, USA) was inserted into the jugular vein. Electrocardiogram and PA and RV pressure were recorded and digitally stored with a sampling frequency of 1000 Hz using a PowerLab data acquisition system (ADInstruments, Sydney, NSW, Australia). Cardiac output (CO) was examined by either the Fick method or thermodilution. Stroke volume (SV) was calculated by CO divided by heart rate (HR). Total arterial compliance was calculated as SV divided by pulse pressure.

### Cardiac MRI: PA flow waves

Cardiac MRI was acquired with a Siemens 1.5-T MR Avanto scanner equipped with a six-element phased-array body coil (Siemens Medical Solutions, Erlangen, Germany). Phase-contrast velocity quantification was performed to obtain volumetric flow curves in the main

PA (Fig. 2*A*). Images of the main PA cross section were obtained with velocity encoding perpendicular to the imaging plane (Fig. 2*A*). The circumference of the main PA was contoured semi-automatically, and the vertebrate or the fat tissue was also contoured manually for the background correction. After subtracting velocity of the background from that in the main PA, velocity of the main PA was multiplied with area of the contour. The sampling period of volumetric flow signals was 45 Hz (22 ms). Volumetric flow signals were interpolated to acquire the signals at 1000 Hz using Mathematica software (Wolfram Research, Champaign, IL, USA) (Fig. 1*B*) (Fukumitsu et al., 2020).

## Cardiac MRI: RV volume, mass and wall tension

RV end-diastolic volume (RVEDV), end-systolic volume (RVESV) and RV mass were obtained by contouring endocardial and epicardial borders of the RV on short-axis at the end-diastolic and end-systolic phase using a medical imaging software (Medis Medical Imaging Systems, Leiden, The Netherlands) (Fukumitsu et al., 2020; Trip et al., 2015; van de Veerdonk et al., 2011). RVEDV, RVESV and RV mass were normalized to body surface area (BSA), respectively. RVEF was calculated as (RVEDV – RVESV)/RVEDV × 100. A change in RV mass/BSA from baseline to follow-up after treatment was determined as

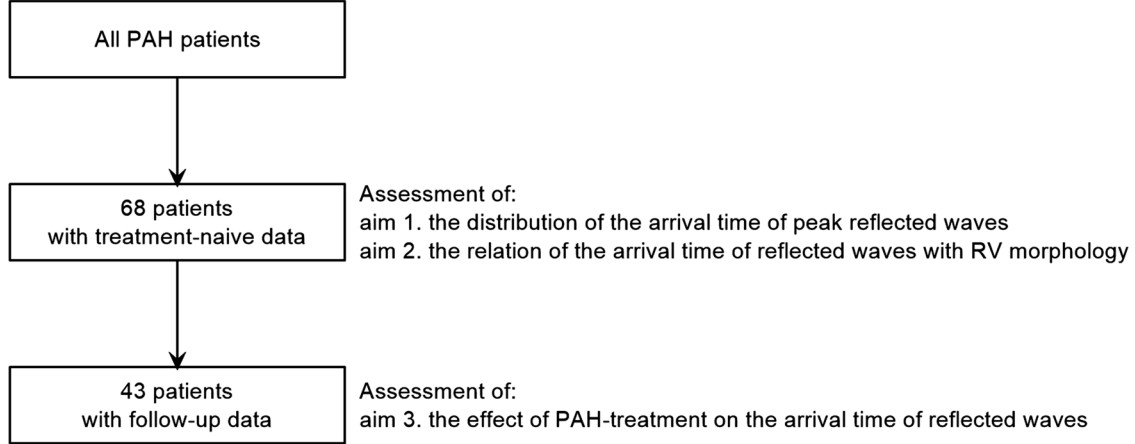

**Figure 1. Study scheme**
Scheme of study population and study aims. RV, right ventricle; PAH, pulmonary arterial hypertension.

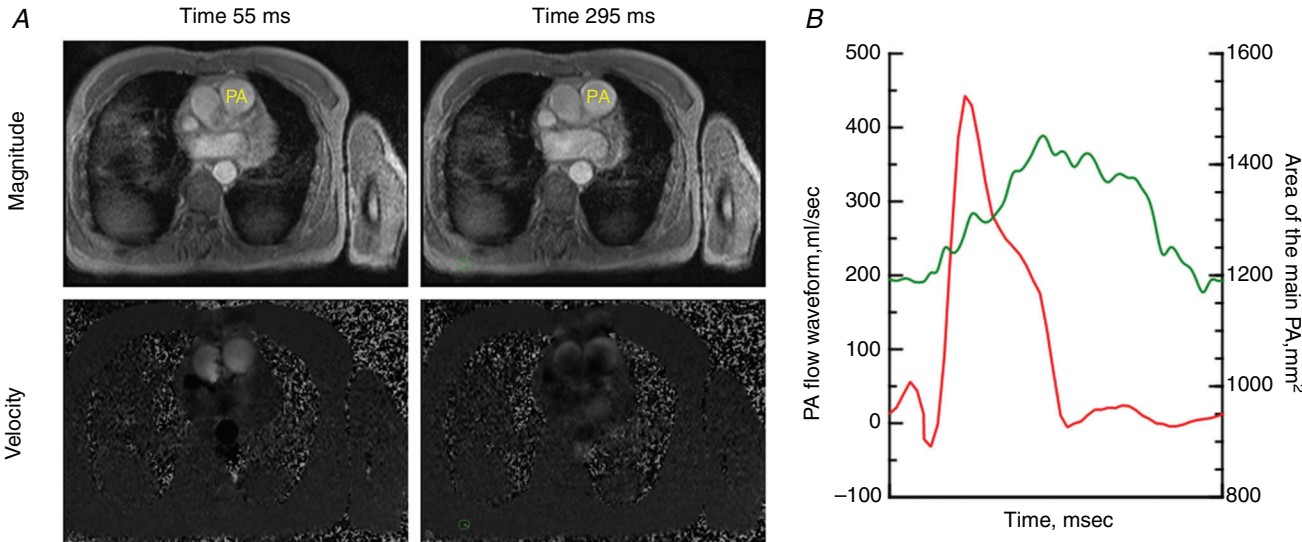

**Figure 2. Cardiac magnetic resonance velocity quantification**
Cardiac magnetic resonance velocity quantification in the main pulmonary artery (PA) *A*, magnitude (top) and velocity (bottom) images of the main PA at two time-points. *B*, curves of PA flow (red) and the area of the main PA (green) during a whole cardiac cycle. [Colour figure can be viewed at wileyonlinelibrary.com]

(RVmass/BSA$_{\text{follow-up}}$-RVmass/BSA$_{\text{baseline}}$)/RVmass/BSA$_{\text{baseline}}$ × 100. PA flow waves during a cardiac cycle were obtained (Fig. 2).

End-diastolic and end-systolic RV wall tension was calculated (Westerhof et al., 2018) as:

$$RV\ wall\ tension\ (t) = RV\ pressure\ (t) \times RV\ radius\ (t) \quad (1)$$

where $t$ is end-diastolic or end-systolic phase. RV radius was estimated assuming that the RV has hemisphere geometry (Fukumitsu et al., 2020; van de Veerdonk et al., 2017) as:

$$RV\ radius\ (t) = \left[ \frac{3}{2\pi} \times RV\ volume\ (t) \right]^{\frac{1}{3}} \quad (2)$$

### Cardiac MRI: compliance of the main PA

The mean area of the main PA was calculated as the averaged value of contour of the main PA over a whole cardiac cycle (Fig. 2B). A change in the area of the main PA during the cardiac cycle ($\Delta$area) was given as difference between the largest and the smallest area. By multiplying $\Delta$ area with 2.0 cm, which is assumed as the length of the main PA for all patients, local compliance of the main PA was given (Saouti et al., 2009) as:

$$Local\ compliance\ of\ the\ main\ PA = \frac{\Delta \text{area} \times 2.0}{SV} \quad (3)$$

### Wave separation analysis

Different cycle length and sampling frequency between PA pressure and flow curves were adjusted as in our prior study (Fukumitsu et al., 2020). PA pressure curves were separated into forward (+) and reflected (−) pressure curves (Westerhof et al., 1972) as:

$$P_{\pm} = \frac{Pm \pm Qm \times Zc}{2} \quad (4)$$

where $Pm$ and $Qm$ were measured pressure and flow curves, respectively, and $Zc$ is characteristic impedance of the proximal PA ($Zc$) calculated as the averaged modulus between the third and tenth harmonics of PA input impedance (Fukumitsu et al., 2020; Nichols et al., 2011). For example, when HR is 90 bpm, the third to tenth harmonics corresponds to 4.5–15 Hz, a frequency range that should be reliably represented by the measured pressure and flow. $Zc$ represents the properties of the proximal arteries between the main PA and reflection sites. With higher stiffness or smaller lumen, $Zc$ becomes larger.

Forward and reflected pressure curves were quantified as peak and pulse pressure. Reflection index was examined (Westerhof et al., 2006) as:

$$Reflection\ index = \frac{PP_-}{PP_+ + PP_-} \quad (5)$$

where $PP+$ is pulse pressure of forward pressure and $PP-$ is the pulse pressure of reflected pressure. Time of peak forward and reflected pressure waves were from the onset of RV systole. The cardiac cycle, especially the duration of RV systole, can affect an interpretation of timing of peak forward and reflected waves. Therefore, time of peak forward and reflected pressure were normalized as the time over the duration of RV systole (% in systole: referred to as time% of peak forward and reflected waves) (Fukumitsu et al., 2020). Specifically, time% of peak reflected waves was given by:

$$Time\%\ of\ peak\ reflected\ waves = \frac{Time\ of\ peak\ reflected\ waves}{Duration\ of\ RV\ systole}$$
$$\times 100 \quad (6)$$

When peak reflected pressure waves returned during diastole, time% of reflected waves was over 100%. The estimated wave speed at the elastic arteries was calculated (Westerhof et al., 2015) using:

$$Wave\ speed = \frac{Zc \times Area}{\rho} \quad (7)$$

where area is the mean area of main PA and $\rho$ is blood density given as 1.04 g cm$^{-3}$.

### Statistical analysis

Continuous variables are presented as the mean $\pm$ SD or median (IQR), as appropriate. Normality was tested by visual assessment of a histogram and a quantile-quantile plot and further confirmed by a Shapiro–Wilk test. One-way ANOVA using Bonferroni's multiple comparison was performed to compare three groups (mid-systolic returners, late-systolic returners and diastolic returners). Variables were log-transformed in case of non-normal distribution. When log-transformed variables were distributed non-normally, a Kruskal–Wallis test with Dunn's multiple comparisons was conducted to compare three groups. To compare variables between baseline and 1 year of follow up, a paired $t$ test or Wilcoxon's signed rank test was applied, where appropriate. Linear regression analysis was applied to assess the relationship of logarithm of time% of peak reflected pressure with RV mass/BSA, following the confirmation that a residual value was distributed normally. $P < 0.05$ was considered statistically significant.

The statistical analyses were performed by using SPSS, version 24 (IBM Corp., Armonk, NY, USA).

## Results

Sixty-eight patients were included in the present study: 84% with idiopathic PAH and 16% with heritable PAH (57% female, 43% male). Figure 3*A* demonstrates the distribution of reflected wave arrival times in the treatment-naïve condition. The peak of the reflected wave appeared in mid-systole in thirty-seven patients (54%), in late systole in 20 patients (29%) and in diastole in 11 patients (16%). When the peak of the reflected waves returned during mid-systole or late-systole, the reflected waves affected the shape of PA flow waves, which was apparent as notched flow waves (Fig. 3*B*).

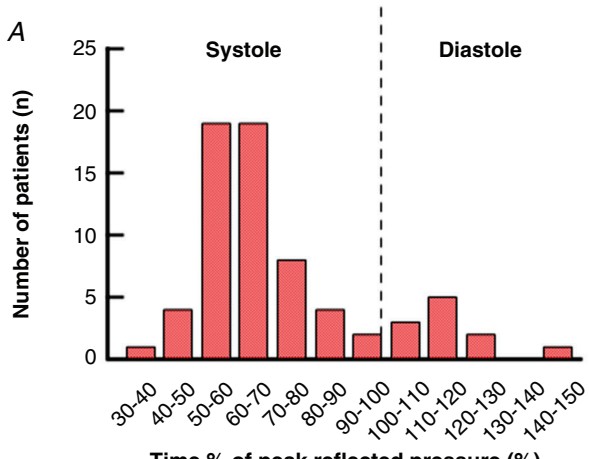

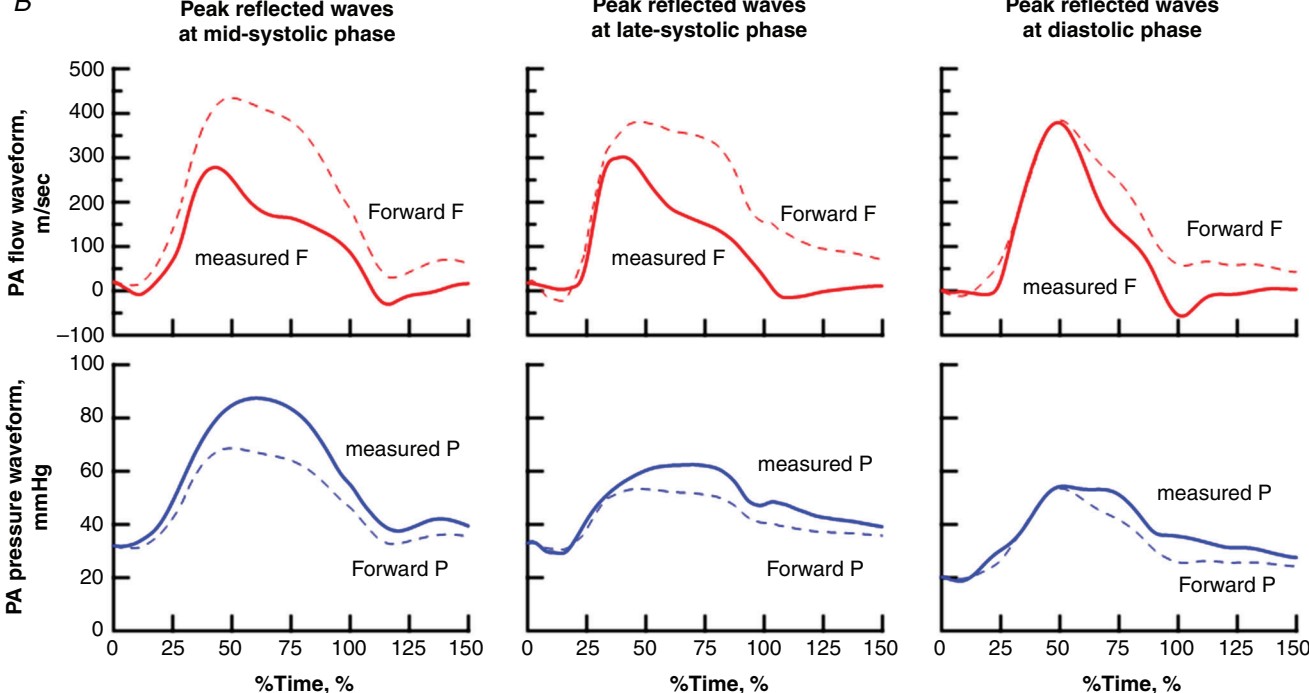

**Figure 3. The distribution of reflected wave arrival times (A) and representative examples (B)**
*A*, distribution of arrival time of peak reflected waves over total study population in the treatment-naive condition (*n* = 68). *B*, representative flow (top) and pressure (bottom) waveforms with peak reflected waves at mid-systolic phase (left), late-systolic phase (middle) and diastolic phase (right). Solid curves present the measured flow or pressure waveform. Dotted curves present forward flow or pressure waveform. Difference between forward and measured waveform present the reflected waveform. The *x*-axis is given by the time of peak reflected wave (% of systole). [Colour figure can be viewed at wileyonlinelibrary.com]

**Table 1. Patient characteristic under treatment-naïve condition (*n* = 68)**

| | Mid-systolic returners (*n* = 37) | Late-systolic returners (*n* = 20) | Diastolic returners (*n* = 11) | *P* value |
|---|---|---|---|---|
| ***Characteristics*** | | | | |
| Age (years) | 59 (30–73) | 58 (43–72) | 64 (50–75) | 0.441 |
| Female (*n*, %) | 23 (62%) | 9 (45%) | 7 (64%) | 0.412 |
| BSA (m², %) | $1.85 \pm 0.21$ | $1.92 \pm 0.21$ | $1.95 \pm 0.24$ | 0.286 |
| PAH type | | | | 0.458 |
| IPAH (*n*, %) | 31 (84%) | 18 (90%) | 8 (73%) | |
| HPAH (*n*, %) | 6 (16%) | 2 (10%) | 3 (27%) | |
| NT-pro BNP (ng L⁻¹) | 1090 (389–2309) | 600 (132–1302) | 626 (253–1231) | 0.097 |
| 6MWD (m) | $396 \pm 135$ (*n* = 29) | $446 \pm 124$ (*n* = 16) | $374 \pm 170$ (*n* = 11) | 0.363 |
| DLco (predict, %) | $55 \pm 18$ (*n* = 30) | $58 \pm 25$ (*n* = 14) | $47 \pm 20$ (*n* = 7) | 0.517 |
| ***Haemodynamics*** | | | | |
| HR (bpm) | $80 \pm 14$ | $76 \pm 15$ | $71 \pm 13$ | 0.174 |
| Mean PAP (mmHg) | 51 (45–58) | 48 (40–54) | 48 (34–50) | 0.964 |
| Pulse pressure (mmHg) | $54 \pm 13$* | $45 \pm 12$ | $45 \pm 11$ | 0.016 |
| Mean RAP (mmHg) | 7 (4–10) | 6 (4–7) | 7 (5–8) | 0.662 |
| CO (L min⁻¹) | 4.1 (3.5–5.6) | 4.7 (4.0–5.9) | 4.4 (3.6–4.9) | 0.528 |
| PVR, Woods unit | 10.0 (7.6–11.9) | 7.3 (6.1–11.5) | 9.1 (4.3–10.6) | 0.031 |
| Total arterial compliance (mL mmHg⁻¹) | 1.0 (0.8–1.4)* | 1.5 (1.0–2.0) | 1.3 (1.1–2.0) | 0.009 |
| *Z*c (mmHg s mL⁻¹) | 0.10 (0.06–0.13)# | 0.14 (0.07–0.24) | 0.18 (0.15–0.19) | 0.006 |
| RVESP (mmHg) | 43 (40–54) | 48 (35–49) | 43 (34–51) | 0.478 |
| RVEDP (mmHg) | 19 (16–23)# | 15 (13–22) | 10 (7–16) | 0.016 |
| Duration of RV systole (ms) | $426 \pm 65$# | $387 \pm 54$ | $335 \pm 55$ | <0.001 |
| ***RV function, morphology and wall tension*** | | | | |
| RVEDV/BSA (mL m⁻²) | 78 (66–97) | 73 (64–85) | 77 (67–88) | 0.440 |
| RVESV/BSA (mL m⁻²) | 49 (39–68) | 42 (34–61) | 43 (34–73) | 0.321 |
| RVEF (%) | $34.8 \pm 11.7$ | $39.6 \pm 10.2$ | $37.2 \pm 15.0$ | 0.349 |
| RV mass/BSA (g m⁻²) | $54.6 \pm 12.6$*# | $44.4 \pm 10.1$ | $42.8 \pm 13.1$ | 0.002 |
| RV-ESWT (kdyne cm⁻¹) | 214 (198–265) | 205 (177–273) | 187 (139–260) | 0.249 |
| RV-EDWT (kdyne cm⁻¹) | 109 (88–125)# | 85 (68–128) | 59 (37–99) | 0.022 |

Data are presented as the mean, mean $\pm$ SD, median (interquartile range) or number of patients (%). *P* value (right column) was by one-way ANOVA or a Kruskal–Wallis test. NT-proBNP, mean PAP, CO, PVR, total arterial compliance, RVEDP, RVEDV/BSA, RVESV/BSA, RV-ESWT and RV-EDWT were log-transformed for one-way ANOVA. There was no statistical difference among groups according to a Kruskal–Wallis test with Dunn's multiple comparisons. *$P < 0.05$: mid-systolic returners *vs.* late-systolic returners, #$P < 0.05$, mid-systolic returners *vs.* diastolic returners.

Abbreviations: PAH, pulmonary arterial hypertension; BSA, body surface area; IPAH, idiopathic PAH; HPAH, heritable PAH; NT-proBNP, N-terminal pro-brain natriuretic peptide; 6MWD, 6 min walk distance; DLco, diffusing capacity of the lungs for carbon monoxide. HR, heart rate; PAP, pulmonary artery pressure; RAP, right atrial pressure; CO, cardiac output; PVR, pulmonary vascular resistance; *Z*c, characteristic impedance of the proximal artery; RVESP, right ventricle end-systolic pressure; RVEDP, right ventricle end-diastolic pressure; RV, right ventricle; RVEDV, right ventricle end-diastolic volume; BSA, body surface area; RVESV, right ventricle end-systolic volume; RVEF, right ventricle ejection fraction; RV-ESWT, right ventricular end-systolic wall tension; RV-EDWT, right ventricular end-diastolic wall tension.

By contrast, when peak reflected waves arrive back during diastole, reflected waves little affected the shape of flow waveform, and flow waves almost maintained a triangular shape.

When comparing mid-systolic, late-systolic and diastolic returners in the treatment-naïve condition, no differences were found in clinical background (similar age, sex, type of PAH and diffusing capacity of the lungs for carbon monoxide) (Table 1). Although HR, mean pulmonary artery pressure (PAP), CO and PVR were also similar, mid-systolic returners were characterized by a higher pulse pressure and a lower total arterial compliance than late-systolic returners, and by a lower *Z*c than diastolic returners (Fig. 4 and Table 1). There were no differences in pulse pressure and total arterial compliance between mid-systolic and diastolic returners. Also, no

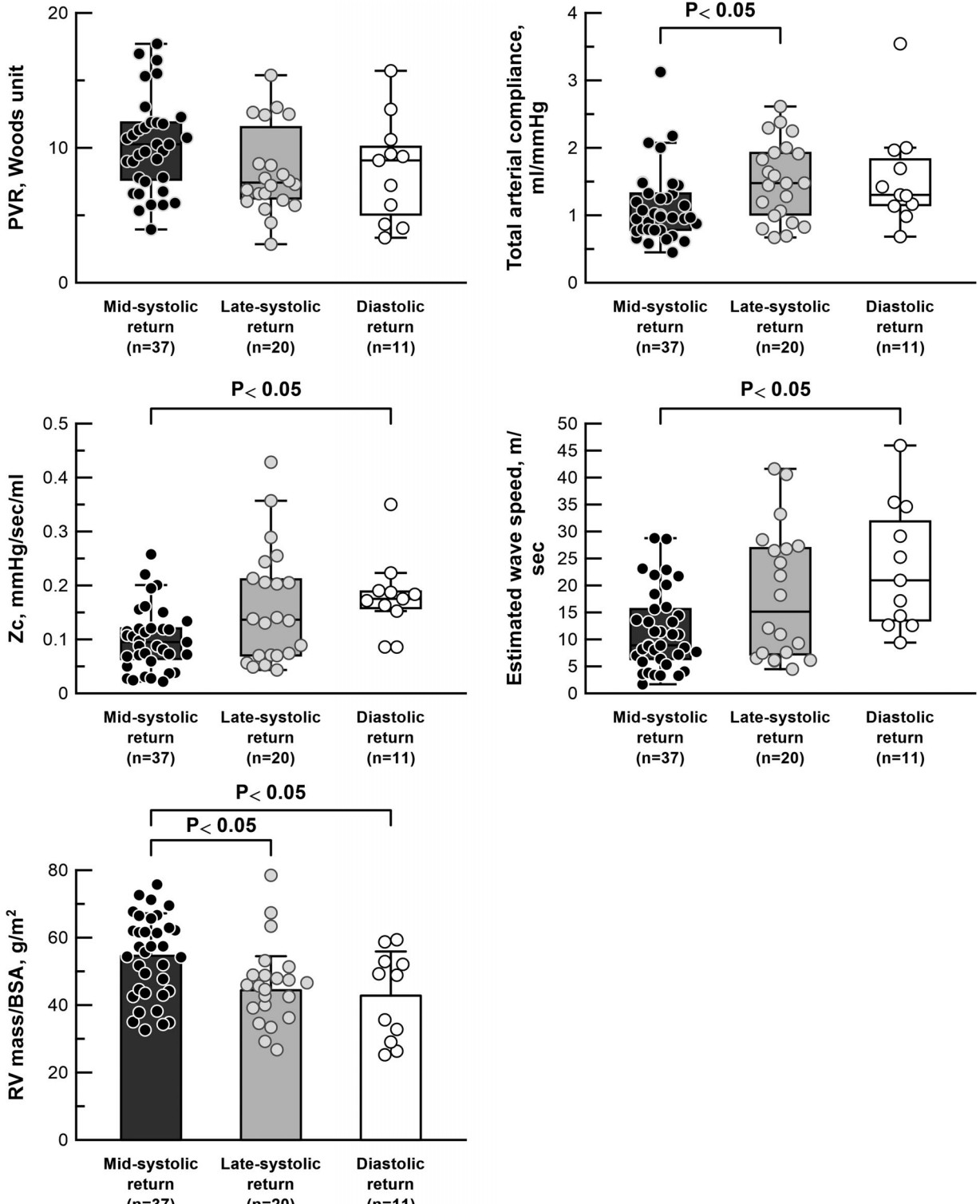

**Figure 4. Right ventricular load and mass in treatment-naïve conditions**
Right ventricular load and mass in treatment-naïve conditions (*n* = 68). Pulmonary vascular resistance (PVR), total arterial compliance and characteristic impedance of the proximal arteries (*Zc*) are represented by a box plot, and RV mass/BSA is represented by a bar chart (mean ± SD). PVR and total arterial compliance were compared using one-way ANOVA after log-transformation. *Zc* was compared using a Kruskal–Wallis test with Dunn's multiple comparison as a result of the non-normal distribution of log-transformed *Zc*. Numbers of patients: 37 in mid-systolic returners, 20 in late-systolic returners and 11 in diastolic returners.

difference was also found in $Zc$ between mid-systolic and late-systolic returners. In mid-systolic returners, higher $Zc$ was associated with lower total arterial compliance and higher pulse pressure, but not in late-systolic or diastolic returners (Fig. 5). RV systolic duration was prolonged more in mid-systolic returners than diastolic returners (Table 1)

In wave separation analysis, the mid-systolic returners had a higher reflection index than diastolic returners (Table 2). Median of time% of peak reflected waves was

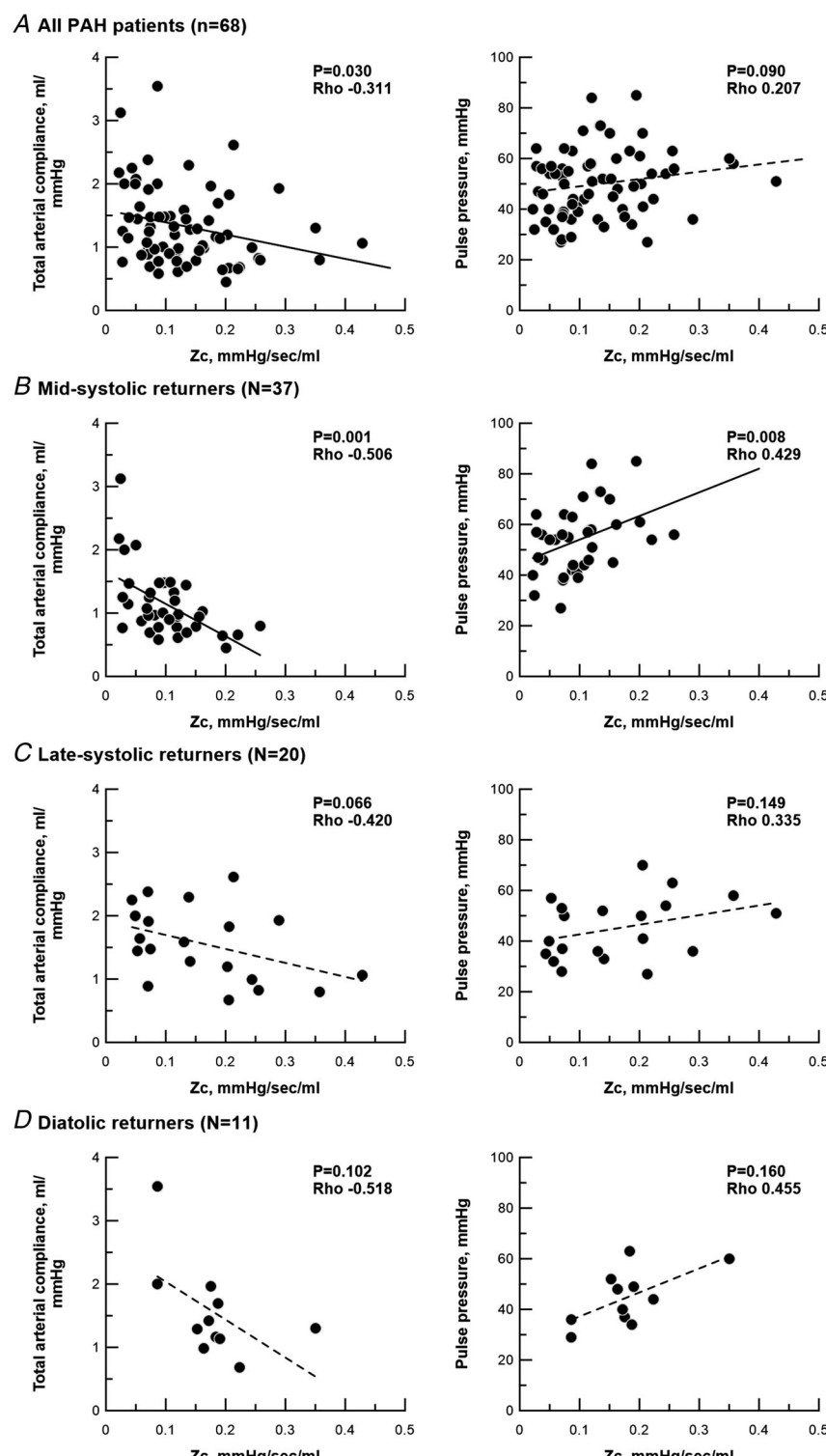

**Figure 5. Characteristic impedance, total arterial compliance and pulse pressure**
Correlations of characteristic impedance with total arterial compliance and pulse pressure. *A*, all PAH patients in the treatment naïve-condition (*n* = 68). *B*, mid-systolic returners (*n* = 37). *C*, late-systolic returners (*n* = 20). *D*, diastolic returners (*n* = 11). Correlations was analysed by a Spearman' rank correlation test. *Zc*, characteristic impedance.

**Table 2. Haemodynamics and wave separation analysis under treatment-naïve condition (*n* = 68)**

| | Mid-systolic returners (*n* = 37) | Late-systolic returners (*n* = 20) | Diastolic returners (*n* = 11) | *P* value |
|---|---|---|---|---|
| **Wave separation analysis** | | | | |
| PP forward (mmHg) | 38 (28–49)[#] | 43 (31–62) | 50 (42–59) | 0.032 |
| PP backward (mmHg) | 20 (17–23) | 21 (13–35) | 20 (13–26) | 0.896 |
| Reflection index | 0.36 (0.31–0.40)[#] | 0.32 (0.28–0.38) | 0.27 (0.25–0.29) | 0.001 |
| Time of forward waves (ms) | 179 (155–226)[#] | 171 (153–201) | 146 (127–171) | 0.039 |
| Time% of forward waves (% of systole) | 44 ± 10 | 46 ± 7 | 44 ± 7 | 0.837 |
| Time of reflected waves (ms) | 242 (213–267)[*#] | 293 (256–346) | 374 (359–409) | <0.001 |
| Time% of reflected waves (% of systole) | 58 (55–63)[*#] | 73 (69–83) | 113 (109–124) | <0.001 |
| **Proximal arteries** | | | | |
| Area of the main PA (mm$^2$) | 864 (756–1017) | 841 (680–1073) | 1003 (766–1306) | 0.445 |
| Compliance of the main PA (mL mmHg$^{-1}$) | 6.2 (4.8–9.9) | 8.0 (4.9–12.8) | 7.8 (6.3–13.7) | 0.497 |
| Estimated wave speed (m s$^{-1}$) | 9 (6–16)[#] | 15 (7–27) | 21 (13–35) | 0.002 |

Data are presented as the mean, mean ± SD or median (interquartile range). *$P < 0.05$ *vs.* late-systolic returners, #$P < 0.05$ *vs.* diastolic returners. *P* value (right column) was by one-way ANOVA or a Kruskal–Wallis test. Mean PAP, CO, PVR, total arterial compliance, reflection index, time forward, time reflection, RV systolic period, area of the main PA, compliance of the main PA and estimated wave speed were log-transformed for one-way ANOVA.

Abbreviations: PAH, pulmonary arterial hypertension; HR, heart rate; PAP, pulmonary artery pressure; RAP, right atrial pressure; CO, cardiac output; PVR, pulmonary vascular resistance; Zc, characteristic impedance of the proximal artery.

58% (% in systole) for mid-systolic returners, 73% for late-systolic returners and 113% for diastolic returners. Properties of the main trunk PA, in terms of mean area and compliance of the main PA, were similar among three groups (Table 2). As a consequence of lower *Zc*, wave transmission in mid-systolic returners was slower than in diastolic returners (mid-systolic returners, 9 m s$^{-1}$ *vs.* diastolic returners, 21 m s$^{-1}$) (Fig. 4 and Table 2).

RV volume and RVEF were similar, however, mid-systolic returners had larger RV mass/BSA than late-systolic and diastolic returners (Fig. 4 and Table 1).

Linear regression analysis indicated that logarithm of time% of peak reflected waves correlated inversely with RV mass/BSA ($r^2 = 0.328$, $P = 0.006$) (Fig. 6). In mid-systolic returners, RV end-diastolic pressure was higher and RV end-diastolic wall tension was excessive (Table 1).

Out of 68 treatment-naïve patients, 43 patients were further examined to perform wave separation analysis at 1 year of follow-up. Duration from baseline to follow-up RHC was 11.9 months (6.1–13.5 months) (Table 3). Thirty-five percent of the patients received monotherapy

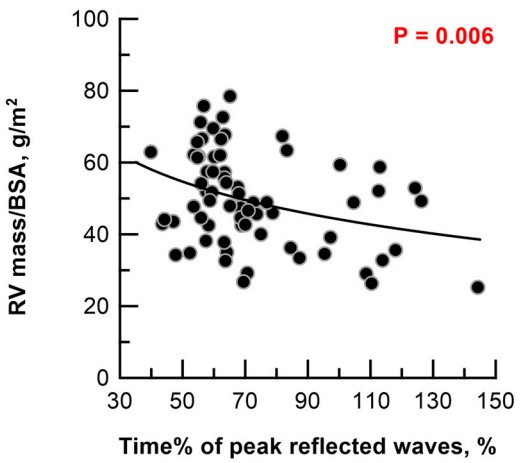
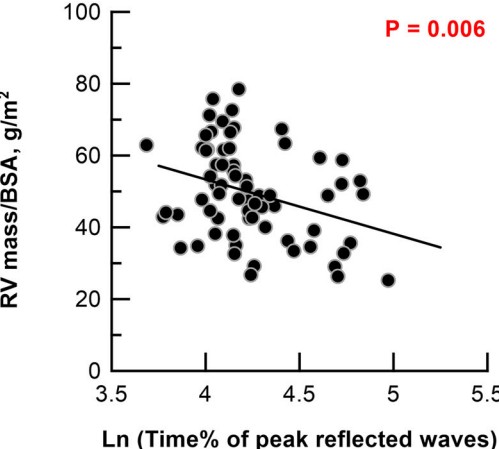

**Figure 6. Linear regression analysis**
Linear regression analysis with logarithm of time of peak reflected waves as the indicator of RV mass/BSA. RV, right ventricle. The *x*-axis is represented by a linear scale. RV mass/BSA = −15 × ln (time% of reflected waves) + 114. The number of analysed patients was 63 for diastolic returners. [Colour figure can be viewed at wileyonlinelibrary.com]

**Table 3. Patient characteristics**

| | Total study population (Aim 1 and 2) | Follow-up population (Aim 3) |
|---|---|---|
| Subjects (*n*) | 68 | 43 |
| Age (years at baseline) | 60.5 (39.4–71.9) | 54.3 (37.1–66.2) |
| Female (*n*, %) | 39 (57%) | 25 (58 %) |
| BSA (m$^2$, %) | 1.89 ± 0.21 | 1.92 ± 0.21 |
| PAH type | | |
| IPAH (*n*, %) | 57 (84%) | 35 (81%) |
| HPAH (*n*, %) | 11 (16%) | 8 (19%) |
| NT-pro BNP (ng L$^{-1}$) | 857 (278–1616) | 845 (290–1545) |
| | (*n* = 57) | (*n* = 38) |
| 6MWD (m) | 406 ± 139 (*n* = 56) | 409 ± 141 (*n* = 28) |
| DLco (predict, %) | 55 ± 20 (*n* = 51) | 59 ± 20 (*n* = 29) |
| Duration from baseline to follow-up (months) | – | 11.9 (6.1–13.5) |
| PAH treatment | | |
| Monotherapy | | |
| PDE5 inhibitor (*n*, %) | – | 7 (16%) |
| ERA (*n*, %) | – | 8 (19%) |
| Combination therapy | | |
| PDE5 inhibitor + ERA (*n*, %) | – | 27 (63%) |
| PDE5 inhibitor + prostacyclin (*n*, %) | – | 1 (2%) |

Total study population was examined for Aims 1 and 2, and follow-up population was analysed for Aim 3.
Abbreviations: BSA, body surface area; PAH, pulmonary arterial hypertension; IPAH, idiopathic PAH; HPAH, heritable PAH; NT-proBNP, N-terminal pro-brain natriuretic peptide; 6MWD, 6-min walk distance; DLco, diffusing capacity of the lungs for carbon monoxide; PDE5 inhibitor, phosphodiesterase inhibitor; ERA, endothelin receptor antagonists.

(16% for PDE5 inhibitor and 19% for ERA) and 65% of patients received combination therapy (63% for PDE5 inhibitor plus ERA and 2% for PDE5 inhibitor plus prostacyclin). PAH treatment decreased HR, mean PAP, and pulse pressure and PVR, and increased total arterial compliance (Table 4). Reflection index was unchanged, although pulse pressure of both forward and reflected waves was attenuated. PAH treatment decreased *Zc* from 0.12 to 0.08 mmHg s mL$^{-1}$, increased compliance of the main PA from 6.8 to 9.1 mL mmHg$^{-1}$, and delayed wave speed from 13.3 to 9.1 m s$^{-1}$ (Fig. 7 and Table 4). As a consequence, time% of peak reflected waves was prolonged from 64% (of systole) to 71%, which means that the return of peak reflected waves was delayed at follow-up (Fig. 8 and Table 4). Decreases in RV end-diastolic pressure and wall tension showed statistical tendency, but were insignificant ($P = 0.08$ for both).

In 30 out of 43 PAH patients, PAH treatment delayed the return of reflected waves. Although clinical background, comprising age, sex and type of PAH treatment, was similar, patients with delayed waves had higher reflection index and earlier return of reflected waves at baseline compared to those with waves that were not-delayed (Table 5). PAH treatment reduced RV hypertrophy in patients with delayed waves, but not in those

with waves that were not-delayed (changes in RV mass; delayed waves *vs.* not-delayed waves, −16 ± 26% *vs.* 2 ± 21%, as % to RV mass/BSA before treatment, $P = 0.036$) (Fig. 8 and Table 5).

## Discussion

We found that, in over half of treatment-naïve PAH patients, the peak reflected wave returned to the RV during mid-systole. Although PVR was similar, patients exhibiting mid-systolic return of the reflected wave had more pronounced RV hypertrophy. PAH treatment attenuated the stiffness of the proximal arteries and decreased wave speed, which delayed the arrival of peak reflected waves. Delayed return of reflected waves was associated with attenuation of RV hypertrophy after treatment.

### Wave reflection as RV load different from PVR

In clinical practice, PVR is the most widely used parameter to assess RV load in PAH. PVR represents 'steady load'; only mean pressures and flows are used in its calculation. On the other hand, the effects of wave reflection can

**Table 4. Measurement at baseline and follow-up (*n* = 43)**

| | Baseline (*n* = 43) | Follow-up (*n* = 43) | *P* value |
|---|---|---|---|
| *Haemodynamics* | | | |
| HR (bpm) | 78 ± 15 | 74 ± 13 | 0.019 |
| Mean PAP (mmHg) | 51 (43–56) | 43 (34–48) | <0.001 |
| Pulse pressure (mmHg) | 52 ± 14 | 42 ± 13 | <0.001 |
| Mean RAP (mmHg) | 6 (4–10) | 7 (4–8) | 0.257 |
| CO (L min$^{-1}$) | 4.5 (3.6–5.4) | 6.1 (4.9–6.9) | <0.001 |
| PVR (Woods unit) | 9.5 (6.6–12.4) | 4.8 (3.7–7.2) | <0.001 |
| Total arterial compliance (mL mmHg$^{-1}$) | 1.13 (0.83–1.48) | 2.04 (1.45–2.72) | <0.001 |
| $Z_c$ (mmHg s mL$^{-1}$) | 0.12 (0.07–0.18) | 0.08 (0.04–0.12) | 0.046 |
| *Wave separation analysis* | | | |
| PP forward (mmHg) | 45 (30–55) | 36 (25–48) | 0.045 |
| PP backward (mmHg) | 21 (16–25) | 18 (14–24) | 0.090 |
| Reflection index | 0.33 ± 0.07 | 0.35 ± 0.06 | 0.251 |
| Time of forward waves (ms) | 189 ± 51 | 212 ± 57 | 0.012 |
| Time% of forward waves (% of systole) | 46.6 ± 9.2 | 50.1 ± 10.4 | 0.059 |
| Time of reflected waves (ms) | 268 (235–321) | 295 (261–401) | 0.010 |
| Time% of reflected waves (% of systole) | 63.7 (57.6–76.9) | 70.7 (63.5–85.1) | 0.017 |
| *Proximal arteries* | | | |
| Area of the main PA (mm$^2$) | 868 (743–1102) | 965 (770–1090) | 0.702 |
| Compliance of the main PA (ml mmHg$^{-1}$) | 6.8 (4.8–13.7) | 9.1 (6.6–14.8) | 0.006 |
| Estimated wave speed (m s$^{-1}$) | 13.3 (6.5–21.8) | 9.1 (4.5–15.6) | 0.041 |
| *RV function and morphology* | | | |
| RVEDV/BSA (mL m$^{-2}$) | 78 (66–87) | 78 (65–90) | 0.496 |
| RVESV/BSA (mL m$^{-2}$) | 46 (38–63) | 40 (32–52) | 0.001 |
| RVEF (%) | 36 ± 12 | 47 ± 10 | <0.001 |
| RV mass/BSA (g m$^{-2}$) | 50 ± 12 | 46 ± 14 | 0.008 |
| *RV wall tension* | | | |
| RV-ESWT (kdyne cm$^{-1}$) | 218 (179–287) | 203 (154–257) | 0.021 |
| RV-EDWT (kdyne cm$^{-1}$) | 99 (68–117) | 76 (52–99) | 0.080 |
| RVESP (mmHg) | 44 (41–54) | 45 (38–56) | 0.114 |
| RVEDP (mmHg) | 17 (13–22) | 15 (10–18) | 0.082 |

Data are presented as the mean, mean ± SD or median (interquartile range).
Abbreviations: PAH, pulmonary arterial hypertension; HR, heart rate; PAP, pulmonary artery pressure; RAP, right atrial pressure; CO, cardiac output; PVR, pulmonary vascular resistance; $Z_c$, characteristic impedance of the proximal artery; RVEDV, right ventricle end-diastolic volume; BSA, body surface area; RVESV, right ventricle end-systolic volume; RVEF, right ventricle ejection fraction; RV-ESWT, right ventricular end-systolic wall tension; RV-EDWT, right ventricular end-diastolic wall tension; RVESP, RV end-systolic pressure; RVEDP, RV end-diastolic pressure.

give insight into the time-resolved effects of the load. Analysis of wave reflection can give us information about RV mechanical force, imposed on a unit length or area of cardiac muscle (units are dyne cm$^{-1}$ for wall tension and dyne cm$^{-2}$ for wall stress) (Fukumitsu et al., 2020). RV wall tension/stress is time-varying, depending on RV size and pressure, and represents the pulsatile load. Thus, wave reflection is a concept describing RV load that is different from PVR. Physiologically, arrival timing of reflected waves can be determined as: (1) wave speed and (2) distance from the RV to the reflection site (Naeije & Huez 2007), both of which are not entirely described by PVR. PVR is the hydraulic resistance, which covers the whole pulmonary vasculature, mainly including small arteries, capillaries and veins.

### The pathophysiology of early return of reflected waves in PAH

Systolic return of reflected waves, implied by a notched shape of the PA flow wave, have been long recognized in PH (Furuno et al., 1991; Kitabatake et al., 1983). Arkles et al. (2011) reported that the notched shape of RV outflow tract flow waves was associated with impaired RV performance in general PH. How mid-systolic reflected waves affect RV performance, however, has remained largely unknown.

In addition to the timing of their arrival, the amplitude of reflected waves should be taken into account when describing RV load. Larger reflected waves during systole can increase RV pressure to a greater extent. In the

present study, patients with a mid-systolic return of the reflected waves exhibited a higher reflection index, which may also contribute to increased RV mass. However, regardless of the amplitude of the reflected wave, their timing is crucial. In our prior CTEPH study, despite the same amplitude of the reflected waves, patients with proximal CTEPH, characterized as earlier return of reflected waves, had higher RV wall stress and ultimately a more deteriorated RV function (Fukumitsu et al., 2020). Pressure augmentation in later-systole or during diastole has little effects on RV mechanical stress (Chirinos et al., 2009).

### Difference of wave reflection between PAH and CTEPH

So far, PAH was considered to be associated with a later return of reflected waves than CTEPH (Castelain et al., 2001; Nakayama et al., 2001). Interestingly, however, the arrival time of reflected waves in PAH was comparable to the subtype of CTEPH that had distal lesions of both lungs (distal CTEPH, mean: 63% of systole) (Fukumitsu et al., 2020). Distal CTEPH and PAH may be indistinguishable in terms of the arrival time of reflected waves. By contrast, the arrival time of reflected waves was not similarly associated with RV structure in PAH and CTEPH. Early return of reflected waves was associated with RV dilatation in CTEPH, but not in

PAH. A possible explanation is that the time course of development of early return of reflected waves may be different between PAH and CTEPH. PAH is characterized by gradual changes in the peripheral pulmonary small arteries (Chan & Loscalzo, 2008), whereas CTEPH is induced by increased pressure after pulmonary embolism in the relatively proximal arteries (Hoeper et al., 2006). Animal experiments indicated that larger (more proximal) vessels were pathologically involved as PAH progressed (Toba et al., 2014). In PAH, therefore, early return of reflected waves may gradually develop with pathological progression. With larger wall thickness, mechanical stress can be more attenuated (Norton, 2001). Thus, the RV in PAH may adapt to increased wall tension by developing hypertrophy. Su et al. (2017) suggested that there were differences in the efficiency of the forward transmission of the waves between PAH and CTEPH, which may reveal the dissimilarities in the RV adaptation to after-load between two diseases. Other possible explanations for the observed differences in relationships of reflected wave to RV shape between PAH and CTEPH may be differences in age or sex distributions, and the more frequent presence of comorbidities in CTEPH. When assessing sex differences in our PAH cohort under treatment naïve-condition (female, $n = 39$; male, $n = 29$), females had faster wave transmission [estimated wave speed; female, 14.4 (9.9–24.2) m s$^{-1}$ vs. male, 8.0 (6.1–16.0) m s$^{-1}$, $P = 0.016$] as a consequence of higher

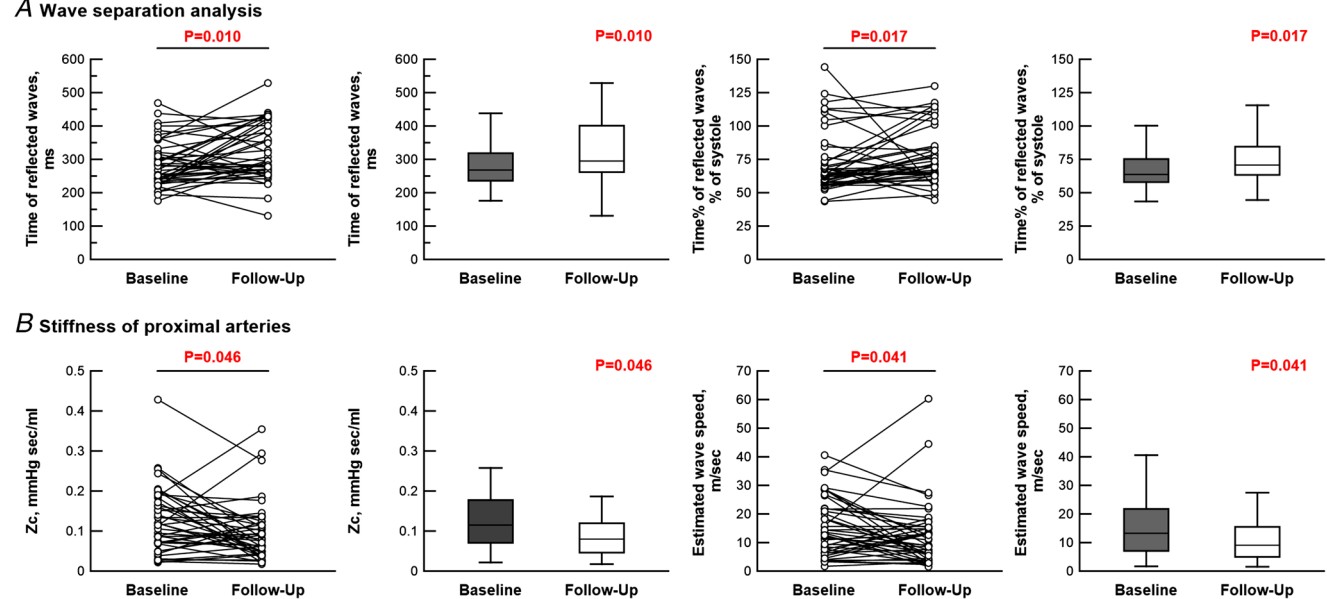

**Figure 7. Wave separation analysis and stiffness of proximal arteries**
Wave separation analysis (*A*) and stiffness of proximal arteries (*B*) at baseline and 1 year of follow-up (*n* = 43). *A*, changes in time of reflected waves (left) and time% of reflected waves (right). *B*, changes in characteristic impedance (*Zc*) (left) and estimated wave speed (right panels). For each, both individual changes and box plot are provided. [Colour figure can be viewed at wileyonlinelibrary.com]

**Table 5. Patient characteristics at baseline in patients with delayed waves and not-delayed waves after PAH treatment**

| | Delayed waves | Not-delayed waves | P value |
|---|---|---|---|
| Subjects, n | 30 | 13 | |
| Age (years at baseline) | 50 (33–65) | 64 (41–71) | 0.300 |
| Female (n, %) | 17 (57%) | 8 (62%) | 1.000 |
| **PAH treatment** | | | |
| Monotherapy | | | |
| PDE5 inhibitor (n, %) | 3 (10%) | 4 (31%) | 0.172 |
| ERA (n, %) | 6 (20%) | 2 (15%) | 1.000 |
| Combination therapy | | | |
| PDE5 inhibitor + ERA (n, %) | 20 (67%) | 7 (54%) | 0.502 |
| PDE5 inhibitor + prostacyclin (n, %) | 1 (3%) | 0 (0%) | 1.000 |
| **Hemodynamics at baseline** | | | |
| HR (bpm) | 72 ± 13 | 70 ± 13 | 0.667 |
| Mean PAP (mmHg) | 52 (44–56) | 50 (47–53) | 0.804 |
| Pulse pressure (mmHg) | 52 ± 15 | 52 ± 13 | 0.969 |
| Mean RAP (mmHg) | 5 (4–8) | 10 (6–11) | 0.054 |
| CO (L min$^{-1}$) | 4.5 (3.6–5.4) | 4.5 (3.9–5.2) | 0.804 |
| PVR (Woods unit) | 10.4 (6.6–12.3) | 9.4 (6.2–12.8) | 0.927 |
| Total arterial compliance (mL mmHg$^{-1}$) | 1.1 (0.9–1.7) | 1.1 (0.8–1.4) | 0.784 |
| Zc (mmHg s mL$^{-1}$) | 0.09 (0.05–0.15) | 0.18 (0.11–0.22) | 0.003 |
| **Wave separation analysis at baseline** | | | |
| PP forward (mmHg) | 39 (26–50) | 55 (35–61) | 0.013 |
| PP backward (mmHg) | 21 (16–25) | 21 (17–24) | 1.000 |
| Reflection index | 0.35 (0.30–0.39) | 0.28 (0.26–0.35) | 0.010 |
| Time of forward wave (ms) | 200 ± 50 | 163 ± 44 | 0.025 |
| Time% of forward waves (% of systole) | 47 ± 10 | 44 ± 8 | 0.326 |
| Time of reflected waves (ms) | 262 (240–302) | 310 (224–394) | 0.366 |
| Time% of reflected waves (% of systole) | 61 (56–69) | 77 (66–111) | 0.003 |
| **Proximal arteries at baseline** | | | |
| Area of the main PA (mm$^2$) | 923 (808–1107) | 812 (687–1141) | 0.202 |
| Compliance of the main PA (mL mmHg$^{-1}$) | 11.8 (7.3–22.0) | 10.0 (7.3–12.0) | 0.380 |
| Estimated wave speed (m s$^{-1}$) | 11 (6–20) | 18 (12–28) | 0,058 |
| **RV function and morphology at baseline** | | | |
| RVEDV/BSA (mL m$^{-2}$) | 77 (66–87) | 79 (69–95) | 0.629 |
| RVESV/BSA (mL m$^{-2}$) | 44 (38–62) | 50 (40–70) | 0.505 |
| RVEF (%) | 36 ± 13 | 34 ± 11 | 0.605 |
| RV mass/BSA (g m$^{-2}$) | 51 ± 11 | 47 ± 16 | 0.248 |
| **RV wall tension at baseline** | | | |
| RV-ESWT (kdyne cm$^{-1}$) | 215 (183–271) | 246 (168–323) | 0.845 |
| RV-EDWT (kdyne cm$^{-1}$) | 94 (67–111) | 112 (70–147) | 0.232 |
| RVESP (mmHg) | 44 (41–54) | 43 (36–64) | 0.804 |
| RVEDP (mmHg) | 17 (12−21) | 19 (14–24) | 0.276 |

Abbreviations: Delayed waves, patients with delayed reflected waves by PAH treatment; Not-delayed waves, patients without delayed reflected waves by PAH treatment; PAH, pulmonary arterial hypertension; PDE5 inhibitor, phosphodiesterase inhibitor, ERA, endothelin receptor antagonists; HR, heart rate; PAP, pulmonary artery pressure; RAP, right atrial pressure; CO, cardiac output; PVR, pulmonary vascular resistance; Zc, characteristic impedance of the proximal artery; PP, pulse pressure; PA, pulmonary artery; RVEDV, right ventricle end-diastolic volume; BSA, body surface area; RVESV, right ventricle end-systolic volume; RVEF, right ventricle ejection fraction; RV-ESWT, right ventricular end-systolic wall tension; RV-EDWT, right ventricular end-diastolic wall tension; RVESP, RV end-systolic pressure; RVEDP, RV end-diastolic pressure.

Zc [female, 0.13 (0.09–0.19) mmHg s mL$^{-1}$ *vs.* male, 0.09 (0.06–0.15) mmHg s mL$^{-1}$, P = 0.039]. There was no sex difference in the mean area of the main PA, suggesting that the main PA was stiffer in female. Although PVR, total arterial compliance, the magnitude and timing of reflected waves, and RV volume and mass were similar, RVEDP was higher in female than male group [female, 19 (14–23) mmHg *vs.* male 16 (12–18) mmHg, P = 0.032). These sex differences may explain the difference of RV adaptation to increased afterload among diseases.

## Effect of PAH treatment on wave speed and the arrival time of reflected waves

Early return of reflected waves can occur (1) when the speed of wave transmission is high or (2) when the distance from the RV to the reflection sites is short (Naeije & Huez 2007). In terms of mechanics, the speed of wave transmission should be considered to be independent of the distance to the reflection sites. Compared with normal subjects, the wave transmits more quickly in PAH as a result of higher $Zc$ (Su et al., 2017). Surprisingly, however, when wave speed was estimated using $Zc$, the area of the main proximal PA and blood density [see eqn (7)], mid-systolic returners had slower wave transmission as a consequence of lower $Zc$ [eqn (4)] than late-systolic or diastolic returners. Here, lower $Zc$ may represent a larger area but not less stiffness of the proximal arteries (see section below for details). Furthermore, the geometry of the distal part of the proximal PA may be different from the mid-systolic returners and late-systolic/diastolic returners (see section below for details). A wider proximal PA with less tapering but abrupt narrowing may induce early return of reflected waves even at a low wave speed. The initial development of PAH may involve the acceleration of wave speed (Su et al., 2017). With the progress of PAH, wave reflection may occur at closer proximity in pulmonary arteries (Toba et al., 2014). Once PAH is developed, the reflection sites rather than wave speed may be more important to cause early return of reflected wave. Further clinical investigations are necessary to confirm this explanation.

Regardless, delaying the arrival of reflected waves may constitute a legitimate goal of PAH treatment. In the present study, PAH treatment was associated with attenuated proximal arterial stiffness (decreased $Zc$ and increased compliance of the main PA) and wave speed, which delayed the arrival of peak reflected waves. As PAH treatment decreased PVR (Table 4), distal vascular vasodilatation could occur, which indicates that the reflection sites moved to more distal locations. RV hypertrophy was reduced when the arrival of reflected waves was delayed, whereas RV hypertrophy persisted when waves were not delayed.

## Lower total compliance and lower characteristic impedance of mid-systolic returners

Mid-systolic returners were characterized by lower total arterial compliance and lower $Zc$. Although total arterial compliance is distributed to the whole pulmonary vasculature (Saouti et al., 2009), $Zc$ is determined by the mechanical properties of the local elastic arteries between the proximal PA and the reflection site (Nichols et al., 2011). $Zc$ is decreased when the radius of elastic arteries becomes large and/or stiffness of elastic arteries is attenuated (Wang & Chesler, 2011). Lower total compliance but lower $Zc$ apparently conflict, and thus should be interpreted carefully.

To explain lower total arterial compliance and lower $Zc$, we should keep in mind that the elastic arteries are not like uniform tubes but gradually taper toward the periphery (Westerhof & Westerhof 2012). When the elastic arteries are separated into a proximal and a distal part, the proximal part is unlikely to explain lower both total arterial compliance and $Zc$ because the properties of the proximal arteries (local area and compliance of the main PA trunk) do not differ (Table 2). Thus, the distal part of elastic arteries is a possible determinant of lower both total arterial compliance and $Zc$. Lower total arterial compliance as well as $Zc$ can be the case if patients have less tapering (larger radius) but stiffer vessels at distal part of elastic arteries (Wang & Chesler, 2011), which may be explain the apparently conflicting results for $Zc$ and compliance. Not gradual tapering, but the abrupt narrowing or obstruction could cause wave reflection in PAH. Further investigations are needed to examine the properties of the distal part of elastic arteries.

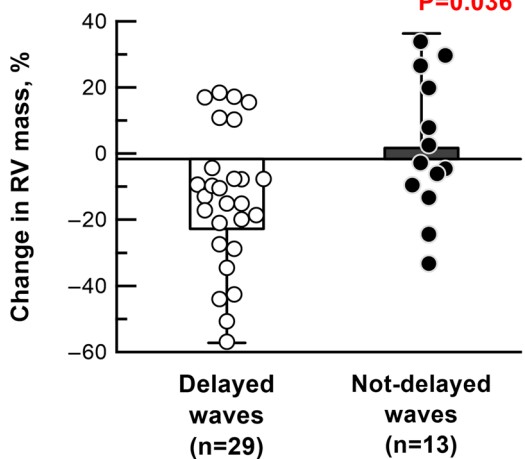

**Figure 8. Change in RV mass after PAH treatment**
Change in RV mass in patients with delayed waves (*n* = 29) and not-delayed waves (*n* = 10) after PAH treatment. Change in RV mass (%) was determined as: (RV mass/BSA$_{follow-up}$ − RV mass/BSA$_{baseline}$)/RV mass/BSA$_{baseline}$ ×100. One patient with delayed waves was excluded as a result of the unavailability of RV mass/BSA at follow-up. [Colour figure can be viewed at wileyonlinelibrary.com]

## Limitations

First, PA flow and pressure waveforms were recorded non-simultaneously. However, the methodology of data processing using non-simultaneous measurements of flow and pressure curves has been already established (Lankhaar et al., 2008; Quail et al., 2014; Saouti et al., 2010;

Schäfer et al., 2018). Cardiac MRI was performed within a median of 1 days before or after RHC. A median difference in HR between RHC and CMRI was 2 bpm (IQR −3 to 7 bpm).

Second, the calculation of RV wall tension was based on the simplification that the RV shape was semi-sphere in PH. However, the RV dilates with leftward septal bowing (Marcus et al., 2008; van de Veerdonk et al., 2017) and the shape of the RV can be considered to be a semi-sphere. Indeed, with this assumption, other research groups already examined RV volume using a conductance catheter method (Tedford et al., 2013; Tello, Dalmer et al., 2019; Tello, Wan et al., 2019; Tello et al., 2020).

Third, wave separation analysis was conducted in the present study, whereas there are several methods for examining wave speed (Mynard et al., 2008) and the return time of reflected waves, including wave intensity analysis (Su et al., 2017). Regardless of the method for examining the return timing of the waves, however, early return of reflected waves can be considered as the determinant of RV mechanical stress in PAH. As already known in the clinical setting, mid-systolic notch of flow waves in echocardiography, which could be induced by mid-systolic return of reflected waves, was associated with RV function in PH (Arkles et al., 2011).

## Conclusions

In the present study, we showed that timing of reflected waves was variable in PAH. Mid-systolic return of reflected waves impacts RV hypertrophy and RV end-diastolic wall tension. Timing of reflected waves is a parameter to be considered in the investigation of RV failure in PAH. PAH treatment acts favourably on the right ventricle not only by reducing PVR, but also by delaying the return of reflected waves.

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

## Additional information

### Data availability statement

The data that support the findings of the study are available from the first and/or corresponding authors upon reasonable request.

### Competing interests

Dr N. J. Braams received a research grant from Actelion Pharmaceuticals. Dr J. T. Marcus received fees as a consultant for Actelion Pharmaceuticals. The remaining authors declare that they have no competing interests.

### Authors contributions

All authors contributed to conception and/or design of the work. MF, JAG, NB and AAB contributed to acquisition of the data. MF and BEW performed by data analysis using Mathematica software (https://www.wolfram.com/mathematica). All authors contributed to interpretation of the data. All authors were involved in drafting the work and approved the final version of the manuscript submitted for publication. All authors agreed to be accountable for all aspects of the work in ensuring that questions related to the accuracy or integrity of any part of the work are appropriately investigated and resolved. All those who qualify for the authorship are listed.

### Funding

Dr M. Fukumitsu was supported by European Respiratory Society Short-Term Research Fellowship 2019 (STRF201904-00595) and Japan Heart Foundation/Bayer Yakuhin Research Grant Abroad. Drs B. E. Westerhof and J. A. Groeneveldt were supported by NWO-VICI (918.16.610). Dr A. Bayoumy was supported by the Egyptian ministry of Higher education. Dr H-J Bogaard was supported by The Netherlands CardioVascular Research Initiative (CVON-2012-08 PHAEDRA, CVON-2017-10 DOLPHIN-GENESIS). Drs A. Vonk Noordegraaf and F. S. de Man were supported by The Netherlands CardioVascular Research Initiative (CVON-2012-08 PHAEDRA, CVON-2017-10 DOLPHIN-GENESIS) and The Netherlands Organization for Scientific Research (NWO-VIDI: 917.18.338, NWO-VICI: 918.16.610).

### Keywords

pulmonary artery–right ventricle coupling, pulmonary hypertension, right ventricle failure, ventricular afterload, wave reflection

## Supporting information

Additional supporting information can be found online in the Supporting Information section at the end of the HTML view of the article. Supporting information files available:

**Statistical Summary Document**
**Peer Review History**

