## [Peer Review History · The Journal of Physiology]

When right ventricular pressure meets volume: the impact of arrival time of reflected waves on right ventricle load in pulmonary arterial hypertension

Masafumi Fukumitsu, Joanne A Groeneveldt, Natalia J. Braams, Ahmed A Bayoumy, J. Tim Marcus, Lilian J. Meijboom, Frances S de Man, Harm-Jan Bogaard, Anton Vonk Noordegraaf, and Berend E. Westerhof
DOI: 10.1113/JP282422

Corresponding author(s): Berend Westerhof (b.e.westerhof@amsterdamumc.nl)

The following individual(s) involved in review of this submission have agreed to reveal their identity: NC Chesler (Referee #1); Ken D O'Halloran (Referee #3)

Review Timeline:

Submission Date:	26-Oct-2021
Editorial Decision:	01-Dec-2021
Revision Received:	28-Dec-2021
Editorial Decision:	04-Mar-2022
Revision Received:	15-Mar-2022
Accepted:	11-Apr-2022

Senior Editor: Bjorn Knollmann

Reviewing Editor: Larissa Shimoda

Transaction Report:

Dear Dr Westerhof,

Re: JP-RP-2021-282422 "When right ventricular pressure meets volume: the impact of arrival time of reflected waves on right ventricle load in pulmonary arterial hypertension" by Masafumi Fukumitsu, Joanne A Groeneveldt, Natalia J. Braams, Ahmed A Bayoumy, J. Tim Marcus, Lilian J. Meijboom, Frances S de Man, Harm-Jan Bogaard, Anton Vonk Noordegraaf, and Berend E Westerhof

Thank you for submitting your manuscript to The Journal of Physiology. It has been assessed by a Reviewing Editor and by 3 expert Referees and I am pleased to tell you that it is considered to be acceptable for publication following satisfactory revision.

The reports are copied at the end of this email. Please address all of the points and incorporate all requested revisions, or explain in your Response to Referees why a change has not been made.

NEW POLICY: In order to improve the transparency of its peer review process The Journal of Physiology publishes online as supporting information the peer review history of all articles accepted for publication. Readers will have access to decision letters, including all Editors' comments and referee reports, for each version of the manuscript and any author responses to peer review comments. Referees can decide whether or not they wish to be named on the peer review history document.

Authors are asked to use The Journal's premium BioRender (<https://biorender.com/>) account to create/redrawn their Abstract Figures. Information on how to access The Journal's premium BioRender account is here: <https://physoc.onlinelibrary.wiley.com/journal/14697793/biorender-access> and authors are expected to use this service. This will enable Authors to download high-resolution versions of their figures.

I hope you will find the comments helpful and have no difficulty returning your revisions within 4 weeks.

Your revised manuscript should be submitted online using the links in Author Tasks Link Not Available.

Any image files uploaded with the previous version are retained on the system. Please ensure you replace or remove all files that have been revised.

REVISION CHECKLIST:

- Article file, including any tables and figure legends, must be in an editable format (eg Word)
- Abstract figure file (see above)
- Statistical Summary Document
- Upload each figure as a separate high quality file
- Upload a full Response to Referees, including a response to any Senior and Reviewing Editor Comments;
- Upload a copy of the manuscript with the changes highlighted.

- A potential 'Cover Art' file for consideration as the Issue's cover image;
- Appropriate Supporting Information (Video, audio or data set https://jp.msubmit.net/cgi-bin/main.plex?form_type=display_requirements#supp).

To create your 'Response to Referees' copy all the reports, including any comments from the Senior and Reviewing Editors, into a Word, or similar, file and respond to each point in colour or CAPITALS and upload this when you submit your revision.

I look forward to receiving your revised submission.

If you have any queries please reply to this email and staff will be happy to assist.

Yours sincerely,

REQUIRED ITEMS:

-The contact information provided for the person responsible for 'Research Governance' at your institution is an author on this paper. Please provide an alternative contact who is not an author on this paper or confirm that the author whose email was provided has sole responsibility for research governance. This is the person who is responsible for regulations, principles and standards of good practice in research carried out at the institution, for instance the ethical treatment of animals, the keeping of proper experimental records or the reporting of results.

-You must start the Methods section with a paragraph headed Ethical Approval. If experiments were conducted on humans confirmation that informed consent was obtained, preferably in writing, that the studies conformed to the standards set by the latest revision of the Declaration of Helsinki, and that the procedures were approved by a properly constituted ethics committee, which should be named, must be included in the article file. If the research study was registered (clause 35 of the Declaration of Helsinki) the registration database should be indicated, otherwise the lack of registration should be noted as an exception (e.g. The study conformed to the standards set by the Declaration of Helsinki, except for registration in a database.). For further information see: <https://physoc.onlinelibrary.wiley.com/hub/human-experiments>

-Your manuscript must include a complete Additional Information section

-The Journal of Physiology funds authors of provisionally accepted papers to use the premium BioRender site to create high resolution schematic figures. Follow this link and enter your details and the manuscript number to create and download figures. Upload these as the figure files for your revised submission. If you choose not to take up this offer we require figures to be of similar quality and resolution. If you are opting out of this service to authors, state this in the Comments section on the Detailed Information page of the submission form.

-Please upload separate high-quality figure files via the submission form.

-Your paper contains Supporting Information of a type that we no longer publish. Any information essential to an understanding of the paper must be included as part of the main manuscript and figures. The only Supporting Information that we publish are video and audio, 3D structures, program codes and large data files. Your revised paper will be returned to you if it does not adhere to our Supporting Information Guidelines

-A Statistical Summary Document, summarising the statistics presented in the manuscript, is required upon revision. It must be on the Journal's template, which can be downloaded from the link in the Statistical Summary Document section here: https://jp.msubmit.net/cgi-bin/main.plex?form_type=display_requirements#statistics

-Papers must comply with the Statistics Policy https://jp.msubmit.net/cgi-bin/main.plex?form_type=display_requirements#statistics

In summary:

-If n {less than or equal to} 30, all data points must be plotted in the figure in a way that reveals their range and distribution. A bar graph with data points overlaid, a box and whisker plot or a violin plot (preferably with data points included) are acceptable formats.

-If $n > 30$, then the entire raw dataset must be made available either as supporting information, or hosted on a not-for-profit

repository e.g. FigShare, with access details provided in the manuscript.

- 'n' clearly defined (e.g. x cells from y slices in z animals) in the Methods. Authors should be mindful of pseudoreplication.

- All relevant 'n' values must be clearly stated in the main text, figures and tables, and the Statistical Summary Document (required upon revision)

- The most appropriate summary statistic (e.g. mean or median and standard deviation) must be used. Standard Error of the Mean (SEM) alone is not permitted.

- Exact p values must be stated. Authors must not use 'greater than' or 'less than'. Exact p values must be stated to three significant figures even when 'no statistical significance' is claimed.

- Statistics Summary Document completed appropriately upon revision

- Please include an Abstract Figure. The Abstract Figure is a piece of artwork designed to give readers an immediate understanding of the research and should summarise the main conclusions. If possible, the image should be easily 'readable' from left to right or top to bottom. It should show the physiological relevance of the manuscript so readers can assess the importance and content of its findings. Abstract Figures should not merely recapitulate other figures in the manuscript. Please try to keep the diagram as simple as possible and without superfluous information that may distract from the main conclusion(s). Abstract Figures must be provided by authors no later than the revised manuscript stage and should be uploaded as a separate file during online submission labelled as File Type 'Abstract Figure'. Please ensure that you include the figure legend in the main article file. All Abstract Figures should be created using BioRender. Authors should use The Journal's premium BioRender account to export high-resolution images. Details on how to use and access the premium account are included as part of this email.

- Author photo and profile. First (or joint first) authors are asked to provide a short biography (no more than 100 words for one author or 150 words in total for joint first authors) and a portrait photograph. These should be uploaded and clearly labelled with the revised version of the manuscript. See Information for Authors for further details.

EDITOR COMMENTS

Reviewing Editor:

This manuscript describes wave separation analysis in a fairly large PAH patient cohort. Wave reflections returning at different time points were associated with differences in impedance and right ventricular characteristics. This type of analysis is underutilized, and while the current study may not provide major mechanistic insights, it does lay a foundation for future work.

The reviewers were in agreement that the methods and conclusions are sound and that this study adds new insights to the field. However, both reviewers also felt that there are important methodological details, further analysis (i.e., separation by sex) and data presentations that should be included to strengthen the impact and interpretation of the data. These requested changes/additions are clearly delineated by the reviewers in their comments and should be easily achievable with the dataset in hand. In particular, since the journal does not have page/figure limitations, I would recommend moving the material in the supplement to the main article document.

For a revision, the authors will need to provide the raw datasets and complete the statistical summary. Please also consider converting the two existing bar graphs (Fig 6C and Fig 4 RV mass) to box and whisker or scatter plots. While it is assumed that "n" refers to the number of subjects for each measurement, it would also be helpful to explicitly define the "n" within the legends or Statistical Analysis section.

Methods Details:

There are a few points that need additional descriptions, as outlined by the reviewers.

Ethics Concerns:

While the study appears to have used previously collected data and was judged not to fall into Human Subjects research by the Institution, it seems likely that informed consent must have been provided at some point for use of the data for research purposes. Please see the comments of the Ethics Editor below.

Senior Editor:

Please ensure that the graphs adhere to JP statistics and data policy.

I concur with the reviewing editor's suggestions. Please ensure that the MS is in compliance with the JP ethics and statistics policies.

REFEREE COMMENTS

Referee #1:

See attached comments.

Referee #2:

In the manuscript, "When right ventricular pressure meets volume", Fukumitsu M et al. used wave separation analysis to investigate the prevalence of mid- and late-systolic return of reflected waves in treatment-naïve patients with PAH. They found that the peak reflected wave returned during mid-systole in over half of the cohort. These participants had more pronounced RV hypertrophy. At follow-up there was decreased stiffness in the proximal arteries accompanied by decreased wave speed and a delay in the arrival of the peak reflected wave.

Major comments:

At baseline where the CMR and RHC performed under treatment naïve conditions? The methods state that the CMR was within 28 days before or after RHC but was treatment started after both the baseline RHC and CMR?

What is time % of peak reflected waves, % of systole? Based on supplemental methods, the time zero is defined as onset of RV systole and then the time forward and reflected were then normalized to the duration of the RV systole? How much variability is there in the length of systole between patients? Did the heart rate of the patient play a factor in the time of return? What is the difference in Delayed and not-delayed waves in Figure 6C and Supplemental table 3?

Figure 5. Did the authors perform linear regression analysis on log-transformed data and then plot it based on non-log transformed data? It might be more appropriate to either show the log-transformed data that was used in the analysis and/or include the equation of the regression line and additional information.

Page 8 and figure 6C. What is the rationale for defining the change in RV mass/BSA from baseline to follow-up as $-(\text{follow-up} - \text{baseline})/\text{follow-up} \times 100$? I would have expected that $(\text{follow-up} - \text{baseline})/\text{baseline} \times 100$ to demonstrate the percent change from baseline.

Supplemental methods: ZC was calculated based on the average modulus between the 3rd and 10th harmonics of the PA input impedance but the PA pressures were measured using fluid-filled catheters. Flow is measured from PA phase contrast velocity quantifications that were sampled at 45Hz and then interpolated to increase sampling rate to 1000Hz. Is there sufficient fidelity to have accurate Impedance measurements at higher harmonics?

Table 1 - What do the * and # symbols represent? Were there any significant pairwise comparisons between groups for PVR?

Table 2 - What is the meaning of lower estimated wave speed in the mid-systolic returners at 9 m/sec compared to the diastolic returners at 21 m/sec? I would have expected a higher wave speed would mean an earlier return of the wave. Is there a disconnect between this measurement of wave speed and sites of reflected waves? It seems that they could be coupled/factors in this measurement.

Minor Comments:

Figure 4. It would be helpful to also see the points in addition to the boxplots to better see the distribution of the data and the number of subjects in each group.

Figure 6 A and B - It would also be helpful to overlay the median or mean with distribution for the time points. It is difficult to appreciate the change from baseline to follow-up.

Referee #3 (ethics review):

Thank you for submitting your manuscript to The Journal of Physiology. There are some issues pertaining to ethics that must be clarified.

It appears that the assessments were made as part of "routine" clinical assessment and that the authors have performed a retrospective analysis based upon the data to hand. If this is the case then this point should be made explicitly in the manuscript. The study was approved by the local ethics committee, and the authors highlight that:

"This study complied with the Declaration of Helsinki. This study did not fall within the scope of the Medical Research Involving Human Subjects Act, as confirmed by the Medical Ethics Review Committee of the VU University Medical Centre (approval No. 2017.025)".

I read this as approval by the ethics committee to use historical clinical data for research purposes. The authors do not include a statement on the issue of informed consent. Is it that the ethics board determined that no informed consent was required, that is, consent that the data collected for clinical purposes could be used for research purpose? I appreciate that this would not have been done prospectively, since the data were not collected as part of a research study (or trial), but rather are now included in a retrospective study. It is understandable given the protracted period covered by the authors (1996-2018) that the requirement for consent might be waived by the committee. This point needs to be clarified by the authors, to include revision of the text in the manuscript, and if so will need to be further considered by The Journal.

END OF COMMENTS

Confidential Review

26-Oct-2021

Summary: Fukumitsu et al. postulate a relationship between right ventricular (RV) wall tension and wave reflections, and use wave separation analysis to examine differences in PAH severity and wave reflection times. The article uses pulmonary artery pressure and flow data from 68 patients with PAH, with follow-up data from 43 of these patients. The authors show that patients with mid-systolic wave reflection times have a significant decrease in characteristic impedance compared to those with reflections returning during diastole, and observe a larger normalized RV mass in patients with mid-systolic wave reflection times vs those with diastolic reflected wave arrival times. The manuscript is well written and draws novel conclusions. However, there are several details that should be included in the manuscript to increase the overall impact. Major and minor comments for revision are provided below.

Major comments:

1. Abstract, page 4: “Wave speed was attenuated from 13.3 to 9.1 m/sec ...” Wave speed values are minimally discussed in this work (results presented in Table 2 and supplement), yet the estimated wave speed plays a critical role in determining characteristic impedance and wave separation. The authors should document differences in wave speed in the main text and include at least one sentence regarding changes in wave speed magnitude for the different wave reflection times.
2. Introduction, page 5: “Furthermore, reflected waves returning during diastole ... have negligible effects on RV pressure and load.” This comment is linked to a prior publication by the group (Fukumitsu et al. 2020) on a proximal and distal CTEPH cohort. On this basis, the conclusion that reflected waves during diastole have “negligible effects” is unwarranted. Either more evidence should be provided or the claim should be deleted.
3. Page 8: “A change in RV mass/BSA from baseline ...” The authors use the ratio

$$\frac{RV/BSA_{follow-up} - RV/BSA_{base}}{RV/BSA_{follow-up}}$$

but the denominator should be the “baseline” value to represent the relative change from baseline.

4. Page 8: To ensure that readers can connect the conclusions in the paper, the authors should include the basic mathematical details of wave separation analysis (equation 3) and wave speed calculation (equation 4). The details in the supplement regarding calculation of Z_c can stay in the supplement. The authors focus on the reflection index, so details on how that is derived would be beneficial in this section.
5. Page 10: “mid-systolic returners were characterized higher pulse pressure and lower total arterial compliance than late-systolic returners, and by a lower Z_c than diastolic returners.” Correlations between pulse pressure, total arterial compliance, and Z_c for all groups could help the reader visualize these relationships better than bar graphs. Or, better yet, a surface map of the dependence of Z_c on these parameters color coded for the different groups (early, mid, and late returners). Overall, the figures accurately present the data but do not provide mechanistic insights.
6. Page 10. It is not clear how compliance of the main PA was assessed. Details should be provided in methods.
7. Figure 4: A boxplot with the wave speed for each reflection type with any statistically significant differences noted should be included.
8. Figure 6 A: It is difficult to interpret these results. Consider plotting a single line from pre- to post-treatment with error bars in addition to each patient trajectory.
9. Page 11: “PAH treatment attenuated the stiffness of the proximal arteries and decreased wave speed...” Can the authors be more specific? Is this based on wave speed data? If so, these quantitative values and test statistics should be provided in the results.
10. Page 11. The authors claim that PVR can be approximated as arterial elastance but in fact arterial elastance is sensitive to both PVR and compliance so this statement should be removed.
11. Page 12. The authors state “Simply, larger reflected waves can increase RV pressure greatly” but this is an oversimplification. Indeed, isn’t the point of this article that the timing of the reflected wave is key to RV pressure and ultimately RV function?

12. Page 13: “A possible explanation is that the time course of development ... may be different between PAH and CTEPH.” The study by Su et. al (doi: 10.1161/JAHA.117.006679) investigated differences in wave transmission in PAH and CTEPH. Could the authors correlate any of their results to this prior study and expand on this possible explanation?
13. Page 13: “Other possible explanations ... may be differences in age or sex distribution.” The authors have nearly equal female/male representation in their cohort. Sex differences in all metrics should be investigated and reported (if none, state that none were found).
14. Page 14: “lower Z_c may represent larger area but not higher stiffness...” Please include this discussion in the main text rather than the supplement; perhaps the authors could consider distal vascular stiffening common in PAH, and link that to these possibly conflicting results in Z_c and compliance?
15. Page 14: “In this study, PAH treatment was associated with attenuated proximal arterial stiffness and wave speed, which delayed the arrival of peak reflected waves.” Are these differences in peak reflected wave arrival times solely the effect of proximal stiffness or could this be due to distal vascular vasodilation?
16. Page 14 The authors should expand their limitations section to include the following two factors. First, how would performing a wave intensity analysis in the time domain provide more or different insight on the relationship between wave propagation and RV wall stress? Second, there are multiple methods for calculating wave speed (e.g., sum of squares method) and wave reflection index (see, e.g., Mynard et al. 10.1016/j.jbiomech.2008.10.002). Would the authors expect differences in methodology to alter the outcomes of this study?
17. Page 15. The conclusions should not depend on data reported in the supplement. If these results are key to the impact of the study, they must be reported in the body of the text.

Minor comments:

1. Page 8. "PA flow waves during a cardiac cycle was obtained" should be "PA flow waves during a cardiac cycle were obtained"
2. Page 9-10: "no differences were found in clinical background (similar age, gender, type ...)" Please replace "gender" with "sex."
3. Page 10: Please include description of the non-significant differences between groups in the text (e.g., Z_c does not show a statistically significant difference between mid-systolic and late-systolic groups).
4. Page 10: For the second to last sentence, should it be "mean PAP" instead of "mean PA"?

December 26th, 2021

Professor Bjorn Knollmann,

Senior Editor, *the Journal of Physiology*

Category: Original Research Article

Title: When right ventricular pressure meets volume: the impact of arrival time of reflected waves on right ventricle load in pulmonary arterial hypertension

We would like to thank all reviewers and ethical reviewers, and the editorial team for your very stimulating comments that helped improving our manuscript. All authors have read the comments carefully and discussed the plan for revision. Our point-by-point responses to all comments from the reviewers are provided.

As the reviewers suggested, we described detailed method of wave reflection, for example, how pressure waves were separated into forward and reflected waves, and how time% of reflected waves was derived, using the specific equations. Furthermore, we provided more details of the ethical considerations in relation to the present study according to the ethical reviewer. A redlined version and a clean version of our revised manuscript are provided.

We hope that the reviewers and the editorial team find the revision satisfactory.

Sincerely yours,

Dr. Berend E Westerhof

Department of Pulmonary Medicine
Amsterdam Universitair Medische Centra,
Vrije Universiteit Amsterdam, Amsterdam Cardiovascular Sciences,
De Boelelaan 1117, 1081HV Amsterdam, The Netherlands
T: Number +31204448113 | F: +31204444328
E: email address: b.e.westerhof@amsterdamumc.nl

Response to Reviewing Editor:

Thank you for your constructive comments and suggestions. We revised our manuscript according to your comments and suggestions.

Comments

This manuscript describes wave separation analysis in a fairly large PAH patient cohort. Wave reflections returning at different time points were associated with differences in impedance and right ventricular characteristics. This type of analysis is underutilized, and while the current study may not provide major mechanistic insights, it does lay a foundation for future work.

The reviewers were in agreement that the methods and conclusions are sound and that this study adds new insights to the field. However, both reviewers also felt that there are important methodological details, further analysis (i.e., separation by sex) and data presentations that should be included to strengthen the impact and interpretation of the data. These requested changes/additions are clearly delineated by the reviewers in their comments and should be easily achievable with the dataset in hand.

Response

We deeply appreciate your comments and suggestions. According to both reviewers' suggestions, we provided the details of method, for example how time% of peak reflected waves was derived in the method section. To clarify this point, we revised the manuscript adding some equations.

Please kindly see the sections of "Cardiac MRI: RV volume, mass and wall tension", "Cardiac MRI: Compliance of the main PA", and "Wave separation analysis".

Response (continued)

We performed further analysis (for example; differences in hemodynamic and wave reflection between female and male, correlations of pulse pressure and total arterial compliance with Z_c in each group in treatment-naïve condition, difference in RV systolic duration among each group).

<Page 16, Lines 366 - 374>

When assessing sex differences in our PAH cohort under treatment naïve-condition (female n=39 and male n=29), females had faster wave transmission [Estimated wave speed; female, 14.4 m/sec (9.9 - 24.2) vs male, 8.0 (6.1 - 16.0), $P=0.016$] as a consequence of higher Z_c [female, 0.13 mmHg/sec/ml (0.09 - 0.19) vs male, 0.09 (0.06 - 0.15), $P=0.039$]. Although PVR, total arterial compliance, the magnitude and timing of reflected waves, and RV volume and mass were similar, RVEDP was higher in female than male [female 19 mmHg (14-23) vs male 16 (12 -18), $P=0.032$]. These sex differences may explain the difference of RV

adaptation to increased afterload among diseases.

<Page 11, Lines 254 - 257>

In mid-systolic returners, higher Z_c was associated with lower total arterial compliance and higher pulse pressure, but not in late-systolic or diastolic returners (Figure 5). RV systolic duration was prolonged more in mid-systolic returners than diastolic returners (Table 1).

Response (further continued)

According to the reviewers' suggestions, we re-created the Figure, for example; by adding the box-plots in Figure 4, providing the correlations of pulse pressure and total arterial compliance with Z_c in each group in Figure 5, adding plots of the log-transformed data in Figure 6, and providing the box-plots in Figure 7.

Please kindly see Figure 4 to 7.

In particular, since the journal does not have page/figure limitations, I would recommend moving the material in the supplement to the main article document.

Response

Thank you very much for your recommendation. We moved the data and discussion in the supplement to the main article document.

For a revision, the authors will need to provide the raw datasets and complete the statistical summary. Please also consider converting the two existing bar graphs [Fig 6C (New Fig 8) and Fig 4 RV mass] to box and whisker or scatter plots. While it is assumed that "n" refers to the number of subjects for each measurement, it would also be helpful to explicitly define the "n" within the legends or Statistical Analysis section.

Response

According to your suggestions, we will submit the raw datasets and the statistical summary. Since data of RV mass/BSA was distributed normally, we created the new Figure with box-plot (mean \pm standard deviations). According to your suggestion, we added the scatter plots in Figure 4 and Figure 8.

Please see Figure 4 and Figure 8.

Response (continued)

In Figure legends, we added the number of patients for each analysis.

Please see Figure legends.

Methods Details:

There are a few points that need additional descriptions, as outlined by the reviewers.

Response

Thank you very much for your comments. We revised the manuscript according to the reviewers' suggestions.

Ethics Concerns:

While the study appears to have used previously collected data and was judged not to fall into Human Subjects research by the Institution, it seems likely that informed consent must have been provided at some point for use of the data for research purposes. Please see the comments of the Ethics Editor below.

Response

We appreciate your suggestions. As we replied to the ethical reviewer, we clarified the ethics in our present study.

<Page 5, Lines 110 - 117>

This study was retrospectively conducted on a registry of patients with diagnosed PAH who routinely underwent right-sided heart catheterization (RHC), cardiac MRI, six-minute walk test and blood sampling. This study complied with the Declaration of Helsinki. Informed consent was given by all patients included after 2010 to use their data for scientific purposes. For patients included before 2010, the medical ethical committee waived the requirement of consenting the patients in retrospect and allowed using the clinical data collected in that period for scientific purposes (approval No. 2017.025).

Response to Senior Editor

Thank you for your constructive comments and suggestions. We revised our manuscript according to your comments and suggestions.

Comments

Please ensure that the graphs adhere to JP statistics and data policy.

Response

Thank you very much. We will submit the raw data and statistical summary according to the policy of *the Journal of Physiology*.

I concur with the reviewing editor's suggestions. Please ensure that the MS in compliance with the JP ethics and statistics policies.

Response

We appreciate your suggestions. According to the reviewing editor and the ethical editor, we revised the manuscript regarding ethics as follows.

<Page 5, Lines 110 - 117>

This study was retrospectively conducted on a registry of patients with diagnosed PAH who routinely underwent right-sided heart catheterization (RHC), cardiac MRI, six-minute walk test and blood sampling. This study complied with the Declaration of Helsinki. Informed consent was given by all patients included after 2010 to use their data for scientific purposes. For patients included before 2010, the medical ethical committee waived the requirement of consenting the patients in retrospect and allowed using the clinical data collected in that period for scientific purposes (approval No. 2017.025).

Response to Referee #1

Thank you for your constructive comments and suggestions. We revised our manuscript according to your comments and suggestions.

Summary: Fukumitsu et al. postulate a relationship between right ventricular (RV) wall tension and wave reflections, and use wave separation analysis to examine differences in PAH severity and wave reflection times. The article uses pulmonary artery pressure and flow data from 68 patients with PAH, with follow-up data from 43 of these patients. The authors show that patients with mid-systolic wave reflection times have a significant decrease in characteristic impedance compared to those with reflections returning during diastole, and observe a larger normalized RV mass in patients with mid-systolic wave reflection times vs those with diastolic reflected wave arrival times. The manuscript is well written and draws novel conclusions. However, there are several details that should be included in the manuscript to increase the overall impact. Major and minor comments for revision are provided below.

Major comments

1. Abstract, page 4: “Wave speed was attenuated from 13.3 to 9.1 m/sec ...” Wave speed values are minimally discussed in this work (results presented in Table 2 and supplement), yet the estimated wave speed plays a critical role in determining characteristic impedance and wave separation. The authors should document differences in wave speed in the main text and include at least one sentence regarding changes in wave speed magnitude for the different wave reflection times.

Response

Thank you for the comments. We agree with you that estimated wave speed is the critical role in considering the timing of returned wave reflection. According to your suggestion, we described the result of estimated wave speed in the main results and the new Table.

<Page 12 Lines 280 - 282>

PAH treatment decreased Z_c from 0.12 to 0.08 mmHg/sec/ml, increased compliance of the main PA from 6.8 to 9.1 ml/mmHg, and delayed wave speed from 13.3 to 9.1 m/sec (Table 4, Figure 7).

Response (continued)

The data of changes in estimated wave speed was included in Table 4 and Figure 7B.

Please kindly see the Table 4 and Figure 7B.

2. Introduction, page 5: “Furthermore, reflected waves returning during diastole ... have negligible

effects on RV pressure and load.” This comment is linked to a prior publication by the group (Fukumitsu et al. 2020) on a proximal and distal CTEPH cohort. On this basis, the conclusion that reflected waves during diastole have “negligible effects” is unwarranted. Either more evidence should be provided or the claim should be deleted.

Response

We appreciate your suggestion very much. We revised this sentence and added a new reference as follows.

<Page 4, Lines 93 - 96>

Furthermore, when reflected waves return during diastole, these waves may not add to RV pressure markedly, since the pulmonary valve is closed, and RV pressure decreases steeply after pulmonary valve closure [Sagawa et al. 1988].

3. Page 8: “A change in RV mass/BSA from baseline ...” The authors use the ratio [(follow-up - baseline)/follow-up], but the denominator should be the “baseline” value to represent the relative change from baseline.

Response

We agree with you thoroughly, and this sentence was mistyped. Correctly, “a change in RV mass/BSA from baseline to follow-up after treatment was determined as $(RV_{mass}/BSA_{follow-up} - RV_{mass}/BSA_{baseline}) / RV_{mass}/BSA_{baseline} \times 100$.” As described in the Figure captions, the value provided in the Figure 8 was calculated as $(RV_{mass}/BSA_{follow-up} - RV_{mass}/BSA_{baseline}) / RV_{mass}/BSA_{baseline}$, and thus the result was unchanged.

<Page 8, Lines 173 - 175>

A change in RV mass/BSA from baseline to follow-up after treatment was determined as $(RV_{mass}/BSA_{follow-up} - RV_{mass}/BSA_{baseline}) / RV_{mass}/BSA_{baseline} \times 100$.

4. Page 8: To ensure that readers can connect the conclusions in the paper, the authors should include the basic mathematical details of wave separation analysis (equation 3: new equation 4) and wave speed calculation (equation 4: new equation 7). The details in the supplement regarding calculation of Zc can stay in the supplement. The authors focus on the reflection index, so details on how that is derived would be beneficial in this section.

Response

We appreciate your suggestions. Based on the journal policy in which the *journal of physiology* does not have page/figure limitations, we described the following mathematical methods in the main text.

<Page 9, Lines 192 - 199>

Different cycle length and sampling frequency between PA pressure and flow curves were adjusted as in our prior study [Fukumitsu M. et al. 2020]. PA pressure curves were separated into forward (+) and reflected (-) pressure curves as follows [Westerhof N. et al. 1972].

$$P_{\pm} = \frac{P_m \pm Q_m \times Z_c}{2} \quad \text{Eq. 4}$$

where P_m and Q_m were measured pressure and flow curves, respectively. Z_c is characteristic impedance of the proximal PA (Z_c) calculated as the averaged modulus between the 3rd to 10th harmonics of PA input impedance [Nichols WW. et al. 2011; Fukumitsu M. et al. 2020].

<Page 10, Lines 214 - 217>

When peak reflected pressure waves returned during diastole, time% of reflected waves was over 100%. The estimated wave speed at the elastic arteries was calculated using the following equation [Westerhof N. et al. 2015].

$$\text{Wave speed} = \frac{Z_c \times \text{Area}}{\rho} \quad \text{Eq. 7}$$

where area is the mean area of main PA and ρ is blood viscosity given as 1.04 g/cm³.

Response (continued)

Furthermore, we clarified the method how the reflection index was derived as follows.

<Page 9, Lines 204 - 213>

Forward and reflected pressure curves were quantified as peak and pulse pressure. Reflection index was examined as follows [Westerhof BE. et al. 2006].

$$\text{Reflection index} = \frac{PP_-}{PP_+ + PP_-} \quad \text{Eq. 5}$$

where PP_+ was pulse pressure of forward pressure, and PP_- was pulse pressure of reflected pressure. Time of peak forward and reflected pressure waves were from the onset of RV systole. The cardiac cycle, especially duration of RV systole, can affect an interpretation of timing of peak forward and reflected waves. Therefore, time of peak forward and reflected pressure were normalized as the time over the duration of RV systole (% in systole: referred to as time% of peak forward and reflected waves) [Fukumitsu M. et al. 2020]. Specifically, time% of peak reflected waves was given by the following equation.

$$\text{Time\% of peak reflected waves} = \frac{\text{Time of peak reflected waves}}{\text{Duration of RV systole}} \times 100 \quad \text{Eq. 6}$$

5. Page 10: “mid-systolic returners were characterized higher pulse pressure and lower total arterial

compliance than late-systolic returners, and by a lower Z_c than diastolic returners.” Correlations between pulse pressure, total arterial compliance, and Z_c for all groups could help the reader visualize these relationships better than bar graphs. Or, better yet, a surface map of the dependence of Z_c on these parameters color coded for the different groups (early, mid, and late returners). Overall, the figures accurately present the data but do not provide mechanistic insights.

Response

Thank you very much. According to your comments, we described the correlations of pulse pressure and total arterial compliance with Z_c for all groups (mid-systolic, late-systolic and diastolic returners) in the main text and Figure 5.

<Page 11, Lines 254 - 257>

In mid-systolic returners, higher Z_c was associated with lower total arterial compliance and higher pulse pressure, but not in late-systolic or diastolic returners (Figure 5)

Please kindly see Figure 5.

6. Page 10. It is not clear how compliance of the main PA was assessed. Details should be provided in methods.

Response

As the reviewer points out, the details of how the compliance of the main PA was calculated were described in the method section as follows.

<Page 8, Lines 187 - 189>

By multiplying $\Delta area$ with 2.0 cm, which is assumed as the length of the main PA for all patients, local compliance of the main PA was given as follows (Saouti N et al. 2009).

$$\text{Local compliance of the main PA} = \frac{\Delta area \times 2.0}{SV} \quad \text{Eq. 3}$$

7. Figure 4: A boxplot with the wave speed for each reflection type with any statistically significant differences noted should be included.

Response

Thank you for your suggestion. We created the new Figure 4 with box plots and statistical significance, where appropriate, of the estimated wave speed in each group (mid-systolic, late-systolic and diastolic returners).

Please kindly see the Figure 4.

8. Figure 6 A (New Figure 7A): It is difficult to interpret these results. Consider plotting a single line from pre- to post-treatment with error bars in addition to each patient trajectory.

Response

According to your suggestion, we added the box plot for each parameter in Figure 7A.

Please see the Figure 7A.

9. Page 11: “PAH treatment attenuated the stiffness of the proximal arteries and decreased wave speed...” Can the authors be more specific? Is this based on wave speed data? If so, these quantitative values and test statistics should be provided in the results.

Response

As the reviewer pointed out, this description is based on the quantitative analysis of estimated wave speed. We clarified the specific numbers as follows.

<Page 12, Lines 280 - 282>

PAH treatment decreased Z_c from 0.12 to 0.08 mmHg/sec/ml, increased compliance of the main PA from 6.8 to 9.1 ml/mmHg, and delayed wave speed from 13.3 to 9.1 m/sec (Table 4, Figure 7).

10. Page 11. The authors claim that PVR can be approximated as arterial elastance but in fact arterial elastance is sensitive to both PVR and compliance so this statement should be removed.

Response

We greatly appreciate your comment. We understand that PVR and arterial elastance should be discussed more carefully. With some assumptions (for example, 1; when end-systolic RV pressure is approximated as mean PAP, 2; mean PAWP is negligible in a disease such as PAH, 3; heart rate remains constant), PVR can be approximated as arterial elastance. To clarify this point, we revised the manuscript as follows.

<Page 13, Lines 307 - 311>

When end-systolic RV pressure is approximated as mean PAP (Trip P. et al. 2013), PVR times HR can be represented as follows.

$$PVR \times HR = \frac{mPAP - mPAWP}{SV} \approx Ea \quad \text{Eq. 8}$$

, where mPAP is mean PAP, and Ea is arterial elastance. In PAH, mPAWP is negligible relative to mPAP. When HR remains constant, therefore, a change in PVR can be approximated as a change in Ea

(=mPAP/SV) in the pressure-volume relationship.

11. Page 12. The authors state “Simply, larger reflected waves can increase RV pressure greatly” but this is an oversimplification. Indeed, isn’t the point of this article that the timing of the reflected wave is key to RV pressure and ultimately RV function?

Response

We agree with the reviewer that the timing of reflected waves is crucial, indeed the main conclusion of our study is about the timing of the reflected waves. On the other hand, when considering the reflected waves, the magnitude of the reflected waves should be also kept in mind. In our prior study of the other disease of PH (CTEPH), despite the same amplitude of reflected waves, earlier return of reflected waves is a key to increase RV stress and deteriorate RV function. To clarify this point, we revised the manuscript as follows.

<Page 14, Lines 331 - 338>

Larger reflected waves during systole can increase RV pressure to a greater extent. In the present study, patients with a mid-systolic return of the reflected waves exhibited a higher reflection index, which may also contribute to increased RV mass. However, regardless of the amplitude of the reflected wave, their timing is crucial. In our prior CTEPH study, despite the same amplitude of the reflected waves, proximal CTEPH, characterized as earlier return of reflected waves, had higher RV wall stress and ultimately more deteriorated RV function (Fukumitsu et al. 2020).

12. Page 13: “A possible explanation is that the time course of development ... may be different between PAH and CTEPH.” The study by Su et. al (doi: 10.1161/JAHA.117.006679) investigated differences in wave transmission in PAH and CTEPH. Could the authors correlate any of their results to this prior study and expand on this possible explanation?

Response

Thank you very much for your suggestions. In addition to the time course of development of reflected waves, Su et al. proposed that the difference in the efficacy of the forward transmission of the waves may affect the difference in the RV adaptation between PAH and CTEPH. We described the possible mechanism in the manuscript as follow.

<Page 15, Lines 352 - 363>

PAH is characterized by gradual changes in the peripheral pulmonary small arteries [Chan SY et al. 2008], while CTEPH is induced by increased pressure after pulmonary embolism in the relatively proximal arteries [Hoepfer MM et al. 2006]. Animal experiments indicated that larger (more proximal) vessels were pathologically involved as PAH progressed [Toba M. et al. 2014]. In PAH, therefore, early return of

reflected waves may gradually develop with pathological progression. With larger wall thickness, mechanical stress can be more attenuated [Norton JM. 2001]. Thus, the RV in PAH may adapt to increased wall tension by developing hypertrophy. Su et al. suggested that there were differences in the efficiency of the forward transmission of the waves between PAH and CTEPH [Su et al. 2017], which may reveal the dissimilarities in the RV adaptation to afterload between two diseases.

13. Page 13: “Other possible explanations ... may be differences in age or sex distribution.” The authors have nearly equal female/male representation in their cohort. Sex differences in all metrics should be investigated and reported (if none, state that none were found).

Response

We appreciate your suggestions very much. We analyzed the difference in hemodynamic and RV function between female and male in treatment-naïve condition. In female, estimated wave speed was higher as a consequence of higher Z_c than male. Although PVR, compliance, the magnitude and timing of reflected waves are similar, RVEDP was higher in female than in male. This difference may contribute to the explanation of the different RV adaptation among diseases.

<Page 16, Lines 366 - 374>

When assessing sex differences in our PAH cohort under treatment naïve-condition (female n=39 and male n=29), females had faster wave transmission [Estimated wave speed; female, 14.4 m/sec (9.9 - 24.2) vs male, 8.0 (6.1 - 16.0), P=0.016] as a consequence of higher Z_c [female, 0.13 mmHg/sec/ml (0.09 - 0.19) vs male, 0.09 (0.06 - 0.15), P=0.039]. Although PVR, total arterial compliance, the magnitude and timing of reflected waves, and RV volume and mass were similar, RVEDP was higher in female than male [female 19 mmHg (14-23) vs male 16 (12 -18), P=0.032]. These sex differences may explain the difference of RV adaptation to increased afterload among diseases.

14. Page 14: “lower Z_c may represent larger area but not higher stiffness...” Please include this discussion in the main text rather than the supplement;

Response

Thank you very much. The section “lower total compliance but also lower characteristic impedance of mid-systolic returners” was included in the main text.

Please kindly see the section “Lower total compliance but also lower characteristic impedance of mid-systolic returners” from lines 406 to 427

<Page 17, Lines 385 - 388>

Here, lower Z_c may represent larger area but not higher stiffness of the proximal arteries (see the details in the section “Lower total compliance but also lower characteristic impedance of mid-systolic returners”)

perhaps the authors could consider distal vascular stiffening common in PAH, and link that to these possibly conflicting results in Z_c and compliance?

Response

As the reviewer pointed out, the distal part of elastic arteries may explain low Z_c as well as low total arterial compliance. As described, less tapering but stiffer vessels at the distal part of the elastic arteries can induce both lower total arterial compliance and lower Z_c . The following explanation was provided in the main discussion.

<Page 18, Lines 422 - 425>

Lower total arterial compliance as well as Z_c can be the case if patients have less tapering (larger radius) but stiffer vessels at distal part of elastic arteries [Wang Z. et al. 2011], which may explain the apparently conflicting result in Z_c and compliance.

15. Page 14: “In this study, PAH treatment was associated with attenuated proximal arterial stiffness and wave speed, which delayed the arrival of peak reflected waves.” Are these differences in peak reflected wave arrival times solely the effect of proximal stiffness or could this be due to distal vascular vasodilation?

Response

PAH treatment can decrease PVR and distal vascular vasodilatation. Therefore, not only decreasing Z_c and wave speed, but also moving the reflection sites to more distal location can delay the arrival of reflected waves after treatment. This point was clarified as follows.

<Page 17, Lines 400 - 402>

As PAH treatment decreased PVR (Table 4), distal vascular vasodilatation could occur, which indicates that the reflection sites moved to more distal locations.

16. Page 14 The authors should expand their limitations section to include the following two factors. First, how would performing a wave intensity analysis in the time domain provide more or different insight on the relationship between wave propagation and RV wall stress? Second, there are multiple methods for calculating wave speed (e.g., sum of squares method) and wave reflection index (see, e.g., Mynard et al. 10.1016/j.jbiomech.2008.10.002). Would the authors expect differences in methodology to alter the outcomes of this study?

Response

We greatly appreciate your comments. As you mentioned, there are several methods to analyze wave transmission, including wave intensity analysis. However, regardless of the methodology to separate wave reflection, early return of reflected waves may be the determinant of RV stress in PAH, because mid-systolic notch of flow waves in echocardiography, which might be caused but mid-systolic return of reflected waves, was associated with poor RV function. According to your suggestions, the limitation was expanded as follows.

<Page 19, Lines 443 - 450>

Third, wave separation analysis was conducted in the present study, while there are several methods to examine wave speed (Mynard et al. 2008) and return time of reflected waves, including wave intensity analysis (Su et al. 2017). Regardless of the method for examining the return timing of the waves, however, early return of reflected waves can be considered as the determinant of RV mechanical stress in PAH. As already known in the clinical settings, mid-systolic notch of flow waves in echocardiography, which could be induced by mid-systolic return of reflected waves, was associated with RV function in PH (Arkles et al. 2011)

17. Page 15. The conclusions should not depend on data reported in the supplement. If these results are key to the impact of the study, they must be reported in the body of the text.

Response

Thank you very much. All data in the conclusion was provided in the main text/Table/Figure.

Minor comments:

1. Page 8. “PA flow waves during a cardiac cycle was obtained” should be “PA flow waves during a cardiac cycle were obtained”

Response

According to your suggestion, I revised the manuscript as follows.

<Page 8, Lines 175 - 176>

PA flow waves during a cardiac cycle were obtained (Figure 2).

2. Page 9-10: “no differences were found in clinical background (similar age, gender, type …)” Please replace “gender” with “sex.”

Response

Thank you. I revised the manuscript as follows.

<Page 11, Lines 246 - 249>

When comparing mid-systolic, late-systolic and diastolic returners in the treatment-naïve condition, no differences were found in clinical background (similar age, sex, type of PAH and diffusing capacity of the lungs for carbon monoxide; Table 1).

3. Page 10: Please include description of the non-significant differences between groups in the text (e.g., Zc does not show a statistically significant difference between mid-systolic and late-systolic groups).

Response

We added the description of non-significance of pulse pressure and total arterial compliance between mid-systolic and diastolic returners, and that of Zc between mid-systolic and late-systolic returners.

<Page 11, Lines 252 - 253>

There were no differences in pulse pressure and total arterial compliance between mid-systolic and diastolic returners. Also, no difference was also found in Zc between mid-systolic and late-systolic returners.

4. Page 10: For the second to last sentence, should it be “mean PAP” instead of “mean PA”

Response

Thank you. We revise the manuscript as follows.

<Page 12, Lines 277 - 278>

PAH treatment decreased HR, mean PAP, and pulse pressure and PVR, and increased total arterial compliance (Table 4).

Response to Referee #2

Thank you for your constructive comments and suggestions. We revised our manuscript according to your comments and suggestions.

In the manuscript, "When right ventricular pressure meets volume", Fukumitsu M et al. used wave separation analysis to investigate the prevalence of mid- and late-systolic return of reflected waves in treatment-naïve patients with PAH. They found that the peak reflected wave returned during mid-systole in over half of the cohort. These participants had more pronounced RV hypertrophy. At follow-up there was decreased stiffness in the proximal arteries accompanied by decreased wave speed and a delay in the arrival of the peak reflected wave.

Major comments

At baseline where the CMR and RHC performed under treatment naïve conditions? The methods state that the CMR was within 28 days before or after RHC but was treatment started after both the baseline RHC and CMR?

Response

We appreciate your comment. As the reviewer pointed out, both CMR and RHC were done in treatment-naïve condition.

<Page 6, Lines 124 - 128>

Thus, enrolment criteria were 1) to have digital data of both PA and RV pressure curves at the initial diagnosis before treatment, and 2) to have PA flow curves obtained by cardiac MRI in treatment-naïve condition within 28 days before or after diagnostic RHC [a median interval between RHC and MRI was 1 day: interquartile range (IQR) 0 – 6 days].

What is time % of peak reflected waves, % of systole? Based on supplemental methods, the time zero is defined as onset of RV systole and then the time forward and reflected were then normalized to the duration of the RV systole?

Response

To clarify the method how time% of peak reflected waves is calculated, we added the description and the equation as follows.

<Page 9, Lines 209 - 213>

Therefore, time of peak forward and reflected pressure were normalized as the time over the duration of RV systole (% in systole: referred to as time% of peak forward and reflected waves) [Fukumitsu M. et al. 2020]. Specifically, time% of peak reflected waves was given by the following equation.

$$\text{Time\% of peak reflected waves} = \frac{\text{Time of peak reflected waves}}{\text{Duration of RV systole}} \times 100 \quad \text{Eq. 6}$$

How much variability is there in the length of systole between patients?

Response

RV systolic duration was prolonged more in mid-systolic returners than diastolic returners. We described this difference in the main manuscript and Table as follows.

<Page 11, Lines 256 - 257>

RV systolic duration was prolonged in mid-systolic returners more than diastolic returners (Table 1).

Please kindly see the Table 1.

Did the heart rate of the patient play a factor in the time of return?

Response

Since the cardiac cycle, especially duration of RV systole, can affect an interpretation of timing of peak forward and reflected waves, we added the explanation in the manuscript as follows.

<Page 9, Lines 208 - 212>

The cardiac cycle, especially duration of RV systole, can affect an interpretation of timing of peak forward and reflected waves. Therefore, time of peak forward and reflected pressure were normalized as the time over the duration of RV systole (% in systole: referred to as time% of peak forward and reflected waves) [Fukumitsu M. et al. 2020].

What is the difference in Delayed and not-delayed waves in Figure 6C (new Figure 8) and Supplemental table 3 (new Table 5)?

Response

Patients with delayed waves had higher reflection index and earlier return of reflected waves at baseline, compared with those with waves that were not-delayed. We clarified this point as follows.

<Page 13, Lines 288 - 291>

Although clinical background. i.e., age, sex and type of PAH treatment, was similar; patients with delayed waves had higher reflection index and earlier return of reflected waves at baseline, compared with those with waves that were not-delayed (Table 5).

Figure 5 (new Figure 6). Did the authors perform linear regression analysis on log-transformed data and then plot it based on non-log transformed data? It might be more appropriate to either show the log-transformed data that was used in the analysis and/or include the equation of the regression line and additional information.

Response

Thank you very much for your suggestion. We created a new figure in which log-transformed data was additionally plotted. The equation of the regression line was similar: $RV\ mass/BSA = -15 \times \ln(\text{time\% of reflected waves}) + 114$.

<Page 30, Lines 695 - 697>

Figure 6. Linear regression analysis with logarithm of time of peak reflected waves as the indicator of RV mass/BSA. RV, right ventricle. X-axis is represented as linear scale. $RV\ mass/BSA = -15 \times \ln(\text{time\% of reflected waves}) + 114$.

Please kindly see the Figure 6.

Page 8 and figure 6C. What is the rationale for defining the change in RV mass/BSA from baseline to follow-up as $-(\text{follow-up} - \text{baseline})/\text{follow-up} \times 100$? I would have expected that $(\text{follow-up} - \text{baseline})/\text{baseline} \times 100$ to demonstrate the percent change from baseline.

Response

We agree with you thoroughly. As we replied to the reviewer #1, this sentence was mistyped. Correctly, “a change in RV mass/BSA from baseline to follow-up after treatment was determined as $(RVmass/BSA_{\text{follow-up}} - RVmass/BSA_{\text{baseline}})/RVmass/BSA_{\text{baseline}} \times 100$.” As described in the Figure captions, the value provided in the Figure 8 was calculated as $(RVmass/BSA_{\text{follow-up}} - RVmass/BSA_{\text{baseline}})/RVmass/BSA_{\text{baseline}}$, and thus the result was unchanged.

<Page 7, Lines 173 - 175>

A change in RV mass/BSA from baseline to follow-up after treatment was determined as $(RVmass/BSA_{\text{follow-up}} - RVmass/BSA_{\text{baseline}})/RVmass/BSA_{\text{baseline}} \times 100$.

Supplemental methods: Zc was calculated based on the average modulus between the 3rd and 10th harmonics of the PA input impedance but the PA pressures were measured using fluid-filled catheters. Flow is measured from PA phase contrast velocity quantifications that were sampled at 45Hz and then interpolated to increase sampling rate to 1000Hz. Is there sufficient fidelity to have accurate Impedance measurements at higher harmonics?

Response

We appreciate your comment. Sampling rates of both RHC and CMRI were sufficient to calculate the impedance at higher harmonics, because for example, when HR is 90 bpm, the 3rd to 10th harmonics corresponds to 4.5 to 15 Hz. For helping the readers' understanding, we added the explanation in the manuscript as follows.

<Page 8, Lines 192 - 201>

Different cycle length and sampling frequency between PA pressure and flow curves were adjusted as in our prior study [Fukumitsu M. et al. 2020]. PA pressure curves were separated into forward (+) and reflected (-) pressure curves as follows [Westerhof N. et al. 1972].

$$P_{\pm} = \frac{Pm \pm Qm \times Zc}{2} \quad \text{Eq. 4}$$

where Pm and Qm were measured pressure and flow curves, respectively. Zc is characteristic impedance of the proximal PA (Zc) calculated as the averaged modulus between the 3rd to 10th harmonics of PA input impedance [Nichols WW. et al. 2011; Fukumitsu M. et al. 2020]. For example, when HR is 90bpm, the 3rd to 10th harmonics corresponds to 4.5 to 15 Hz, a frequency range which should be reliably represented by the measured pressure and flow.

Table 1 - What do the * and # symbols represent? Were there any significant pairwise comparisons between groups for PVR?

Response

Thank you for your comments. A symbol * means P<0.05 between mid-systolic and late-systolic returners. A symbol # means P<0.05 between mid-systolic and diastolic returners. There was no statistical difference among groups when a Kruskal-Wallis test with Dunn's multiple comparisons.

Please kindly see the caption of Table 1.

Table 2 - What is the meaning of lower estimated wave speed in the mid-systolic returners at 9 m/sec compared to the diastolic returners at 21 m/sec? I would have expected a higher wave speed would mean an earlier return of the wave. Is there a disconnect between this measurement of wave speed and sites of reflected waves? It seems that they could be coupled/factors in this measurement.

Response

As the reviewer pointed out, the difference in wave speed between mid-systolic and diastolic returners should be discussed more carefully. The wave speed should be considered to be independent of the distance to the reflection sites, therefore our result implies the possibility that the location of the reflection sites is

more important to induce early return of reflected waves, rather than wave speed. Based on low Z_c and low total arterial compliance, a wider proximal PA with abrupt narrowing may induce early return of reflected waves at a low wave speed. To clarify these points, we added the explanation as follows.

<Page 16, Lines 377 - 380>

Early return of reflected waves can occur 1) when the speed of wave transmission is high or 2) when the distance from the RV to the reflection sites is short [Naeije R. and Huez S. 2007]. *In terms of mechanics, the speed of wave transmission should be considered to be independent of the distance to the reflection sites.*

<Page 17, Lines 388 - 392>

Furthermore, a geometry of the distal part of the proximal PA may be different from the mid-systolic returners and late-systolic/diastolic returners (see the details in the following section). A wider proximal PA with less tapering but abrupt narrowing may induce early return of reflected waves even at a low wave speed.

Minor Comments:

Figure 4. It would be helpful to also see the points in addition to the boxplots to better see the distribution of the data and the number of subjects in each group.

Response

Thank you very much. We created the new Figure 4 with box-plot and the number of patients in each group.

Please kindly see the Figure 4.

Figure 6 A and B (Figure 7A and B) - It would also be helpful to overlay the median or mean with distribution for the time points. It is difficult to appreciate the change from baseline to follow-up.

Response

According to the reviewer's suggestion, we created the new Figure 7 with box-plot.

Please kindly see the Figure 7.

Response to Referee #3 (ethics review)

Thank you for your constructive comments and suggestions. We revised our manuscript according to your comments and suggestions.

Thank you for submitting your manuscript to *The Journal of Physiology*. There are some issues pertaining to ethics that must be clarified.

It appears that the assessments were made as part of "routine" clinical assessment and that the authors have performed a retrospective analysis based upon the data to hand. If this is the case then this point should be made explicitly in the manuscript. The study was approved by the local ethics committee, and the authors highlight that:

"This study complied with the Declaration of Helsinki. This study did not fall within the scope of the Medical Research Involving Human Subjects Act, as confirmed by the Medical Ethics Review Committee of the VU University Medical Centre (approval No. 2017.025)".

I read this as approval by the ethics committee to use historical clinical data for research purposes. The authors do not include a statement on the issue of informed consent. Is it that the ethics board determined that no informed consent was required, that is, consent that the data collected for clinical purposes could be used for research purpose? I appreciate that this would not have been done prospectively, since the data were not collected as part of a research study (or trial), but rather are now included in a retrospective study. It is understandable given the protracted period covered by the authors (1996-2018) that the requirement for consent might be waived by the committee. This point needs to be clarified by the authors, to include revision of the text in the manuscript, and if so will need to be further considered by *The Journal*.

Response

We greatly appreciate your comments and suggestions. The present study was retrospectively conducted from a registry of patients with diagnosed pulmonary arterial hypertension who routinely underwent right-sided heart catheterization and cardiac MRI. Informed consent was given by all patients after 2010 to use their data for scientific purposes. Before 2010, the medical ethical committee waived the requirement of consenting the patients in retrospect and allowed using the clinical data collected in that period for scientific purposes (approval No. 2017.025). To clarify these points, we revised the manuscript as follows.

<Page 5, Lines 110 - 117>

This study was retrospectively conducted on a registry of patients with diagnosed PAH who routinely underwent right-sided heart catheterization (RHC), cardiac MRI, six-minute walk test and blood sampling. This study complied with the Declaration of Helsinki. Informed consent was given by all patients included

after 2010 to use their data for scientific purposes. For patients included before 2010, the medical ethical committee waived the requirement of consenting the patients in retrospect and allowed using the clinical data collected in that period for scientific purposes (approval No. 2017.025).

Dear Dr Westerhof,

Re: JP-RP-2021-282422R1 "When right ventricular pressure meets volume: the impact of arrival time of reflected waves on right ventricle load in pulmonary arterial hypertension" by Masafumi Fukumitsu, Joanne A Groeneveldt, Natalia J. Braams, Ahmed A Bayoumy, J. Tim Marcus, Lilian J. Meijboom, Frances S de Man, Harm-Jan Bogaard, Anton Vonk Noordegraaf, and Berend E. Westerhof

Thank you for submitting your revised Research Article to The Journal of Physiology. It has been assessed by the original Reviewing Editor and Referees and has been well received. Some final revisions have been requested.

The reports are copied at the end of this email. Please address all of the points and incorporate all requested revisions, or explain in your Response to Referees why a change has not been made.

NEW POLICY: In order to improve the transparency of its peer review process The Journal of Physiology publishes online as supporting information the peer review history of all articles accepted for publication. Readers will have access to decision letters, including all Editors' comments and referee reports, for each version of the manuscript and any author responses to peer review comments. Referees can decide whether or not they wish to be named on the peer review history document.

I hope you will find the comments helpful and have no difficulty returning revisions within 2 weeks.

If you need to check to make sure that your Methods section conforms to the principles of UK regulations, you may wish to refer to Grundy (2015):
Grundy (2015) J. Physiol. 2015 Jun 15;593(12):2547-9 <https://doi.org/10.1113/JP270818>

Your revised manuscript should be submitted online using the links in Author Tasks Link Not Available. This link is to the Corresponding Author's own account, if this will cause any problems when submitting the revised version please contact us.

The image files from the previous version are retained on the system. Please ensure you replace or remove any files that have been revised.

REVISION CHECKLIST:

- Summary data must be reported as mean {plus minus} SD or 95% confidence interval
- All table and figure legends with summary data must include the statistical test used in the table/figure and sample size
- Figures with summary data bars must include individual data points, or box whisker plots when $n > 30$.
- Article file, including any tables and figure legends, must be in an editable format (eg Word)
- Upload each figure as a separate high quality file
- Upload a full Response to Referees, including a response to any Senior and Reviewing Editor Comments;
- Upload a copy of the manuscript with the changes highlighted.

- A potential 'Cover Art' file for consideration as the Issue's cover image;
- Appropriate Supporting Information (Video, audio or data set https://jp.msubmit.net/cgi-bin/main.plex?form_type=display_requirements#supp).

To create your 'Response to Referees' copy all the reports, including any comments from the Senior and Reviewing Editors, into a Word, or similar, file and respond to each point in colour or CAPITALS and upload this when you submit your revision.

I look forward to receiving your revised submission.

If you have any queries please reply to this email and the Peer Review Coordinator will be pleased to advise.

If revision is not possible, or if you cannot respond to the requests for change, contact us by return email as soon as possible, giving reasons for the difficulties. Withdrawal of the manuscript may be necessary in these circumstances, and instruction will be given on how to proceed. Please note that a paper must be withdrawn before it can be submitted to another journal. If any issues remain unresolved please contact the Publications Office at jphysiol@physoc.org

If you would like help with English language editing, or other article preparation support, Wiley Editing Services offers expert help with English Language Editing, as well as translation, manuscript formatting, and figure formatting at www.wileyauthors.com/eeo/preparation. You can also check out our resources for Preparing Your Article for general guidance about writing and preparing your manuscript at www.wileyauthors.com/eeo/prepresources.

Yours sincerely,

Bjorn Knollmann
Senior Editor
The Journal of Physiology

REQUIRED ITEMS:

-Papers must comply with the Statistics Policy https://jp.msubmit.net/cgi-bin/main.plex?form_type=display_requirements#statistics

In summary:

-If $n \leq 30$, all data points must be plotted in the figure in a way that reveals their range and distribution. A bar graph with data points overlaid, a box and whisker plot or a violin plot (preferably with data points included) are acceptable formats.

-If $n > 30$, then the entire raw dataset must be made available either as supporting information, or hosted on a not-for-profit repository e.g. FigShare, with access details provided in the manuscript.

- n clearly defined (e.g. x cells from y slices in z animals) in the Methods. Authors should be mindful of pseudoreplication.

-All relevant n values must be clearly stated in the main text, figures and tables, and the Statistical Summary Document (required upon revision)

-The most appropriate summary statistic (e.g. mean or median and standard deviation) must be used. Standard Error of the Mean (SEM) alone is not permitted.

-Exact p values must be stated. Authors must not use 'greater than' or 'less than'. Exact p values must be stated to three significant figures even when 'no statistical significance' is claimed.

-Statistics Summary Document completed appropriately upon revision

EDITOR COMMENTS

Reviewing Editor:

If the Statistical Summary Document has errors please describe what is incorrect?:

It seems that the document is not complete. The data presented in some figures and tables is described, but is missing for others (i.e., Figure 5 and 7).

Comments to the Author:

The authors have revised the manuscript and addressed most of the reviewer questions and the ethics concern. There are a few minor suggestions for revision from one reviewer, and it seems that perhaps the Summary Statistics Table is incomplete. These items should be addressed before publication.

Senior Editor:

Comments to the Author:

The revised MS is much improved and only minor deficiencies remain as detailed by the reviewer and reviewing editor.

REFEREE COMMENTS

Referee #1:

The revised manuscript addresses most critiques, with valuable and insightful information and clarifications included. However, there remain some issues that should be addressed.

Comments:

1. Line 309-310: The authors claim E_a can be approximated by $mPAP/SV$ when PAWP is negligible. Recently, however, Singh et al. stated that $mPAP$ has been frequently found to underestimate ESP in a hypertensive RV and overestimate ESP in a normotensive RV. Furthermore, Tello et al. developed an equation ($ESP = 1.65 \times mPAP - 7.79$) to account for this discrepancy. This should be acknowledged in either the main text or addressed as a potential limitation.

2. Line 367-369: Wave speed and Z_c are statistically different across sex, but the authors do not comment on average PA area. If this variable does not show a statistically significant difference with sex, please state this.

3. Lines 380-387: The authors use Z_c synonymously with stiffness, but state "lower Z_c may represent larger area but not higher stiffness." This contradicts the former statement and needs to be consistent.

4. Lines 392-394: Wave speed would increase if PAH caused further vascular stiffening. Wave speed would contribute, then, to early reflected wave returns in addition to reflection sites. Both wave speed and reflection sites are important. Please clarify this.

Referee #3:

Thank you for the revisions made to the text and for including this important information. All points raised previously have been satisfactorily addressed.

END OF COMMENTS

1st Confidential Review

28-Dec-2021

March 14th, 2022

Professor Bjorn Knollmann,

Senior Editor, *the Journal of Physiology*

Category: Original Research Article

Title: When right ventricular pressure meets volume: the impact of arrival time of reflected waves on right ventricle load in pulmonary arterial hypertension

We would like to thank all reviewers and ethical reviewers, and the editorial team again for your kind review and comments that helped improving our manuscript. We have read the comments carefully and discussed the plan for revision. Our point-by-point responses to all comments from the reviewers are provided.

As the reviewing editor suggested, we updated the Statical Summary Document by adding the statistical information about Figure 5&7. We clarified the several concerns in the main text according to the reviewer #1. A redlined version and a clean version of our revised manuscript are provided.

We hope that the reviewers and the editorial team find the revision satisfactory.

Sincerely yours,

Dr. Berend E Westerhof

Department of Pulmonary Medicine
Amsterdam Universitair Medische Centra,
Vrije Universiteit Amsterdam, Amsterdam Cardiovascular Sciences,
De Boelelaan 1117, 1081HV Amsterdam, The Netherlands
T: Number +31204448113 | F: +31204444328
E: email address: b.e.westerhof@amsterdamumc.nl

Response to Reviewing Editor:

Thank you for your constructive comments and suggestions. We revised our manuscript according to your comments and suggestions.

Comments

If the Statistical Summary Document has errors please describe what is incorrect?: It seems that the document is not complete. The data presented in some figures and tables is described, but is missing for others (i.e., Figure 5 and 7).

Response

We sincerely appreciate your comments regarding the Statistical Summary Document. We provided the statistical information about Figure 5 and 7 in the document.

Please kindly see the Statistical Summary Document.

The authors have revised the manuscript and addressed most of the reviewer questions and the ethics concern. There are a few minor suggestions for revision from one reviewer, and it seems that perhaps the Summary Statistics Table is incomplete. These items should be addressed before publication.

Response

Thank you very much for your constructive comments. In addition to updating the Statistical Summary Document, we revised the manuscript according to the Reviewer #1.

Response to Senior Editor

Thank you for your constructive comments and suggestions.

Comments

The revise MS is much improved and only minor deficiencies remain as detailed by the reviewer and reviewing editor.

Response

We appreciate your kind review and comment. According to the suggestion from the reviewing editor and the reviewer #1, we updated the Statistical Summary Document, and revised the manuscript.

Response to Referee #1

Thank you for your constructive comments and suggestions. We revised our manuscript according to your comments and suggestions.

Comments

The revised manuscript addresses most critiques, with valuable and insightful information and clarifications included. However, there remain some issues that should be addressed.

1. Line 309-310: The authors claim E_a can be approximated by mPAP/SV when PAWP is negligible. Recently, however, Singh et al. stated that mPAP has been frequently found to underestimate ESP in a hypertensive RV and overestimate ESP in a normotensive RV. Furthermore, Tello et al. developed an equation ($ESP = 1.65 \times mPAP - 7.79$) to account for this discrepancy. This should be acknowledged in either the main text or addressed as a potential limitation.

Response

We sincerely appreciate your constructive comments and suggestions. As you pointed out previously (as the major comments #11 in the first revision), the description of arterial elastance could potentially cause confusion, partly because there is an ongoing discussion of whether mPAP is approximated as RVESP or not. Thus, we had a discussion with the team again, and reached the conclusion that the description of arterial elastance is not required in this part. In the revised manuscript, we simply discussed the difference between PVR and RV wall stress/tension, in terms of steady or pulsatile load. Since arterial elastance is not analyzed in this study, we left out the discussion about the calculation of arterial elastance to avoid the reader's confusion. Thank you very much for your kind comments.

<Page 13, Line 304-316>

In clinical practice, PVR is the most widely used parameter to assess RV load in PAH. PVR represents "steady load"; only mean pressures and flows are used in its calculation. On the other hand, the effects of wave reflection can give insight into the time-resolved effects of the load. Analysis of wave reflection can give us information about RV mechanical force, imposed on a unit length or area of cardiac muscle (units are dyne/cm for wall tension and dyne/cm² for wall stress) [Fukumitsu M. et al. 2020]. RV wall tension/stress is time-varying, depending on RV size and pressure, and represents the pulsatile load. Thus, wave reflection is a concept describing RV load that is different from PVR. Physiologically, arrival timing of reflected waves can be determined as 1) wave speed and 2) distance from the RV to the reflection site [Naeije R. and Huez S. 2007], both of which are not entirely described by PVR. PVR is the hydraulic resistance which covers the whole pulmonary vasculature, mainly including small arteries, capillaries, and veins.

2. Line 367-369: Wave speed and Zc are statistically different across sex, but the authors do not comment on average PA area. If this variable does not show a statistically significant difference with sex, please state this.

Response

Thank you for your suggestion. There is no sex difference in the mean area of the main PA. To clarify this point, we revised the manuscript as follows.

<Page 16, Line 361-366>

When assessing sex differences in our PAH cohort under treatment naïve-condition (female n=39 and male n=29), females had faster wave transmission [Estimated wave speed; female, 14.4 m/sec (9.9 - 24.2) vs male, 8.0 (6.1 - 16.0), P=0.016] as a consequence of higher Zc [female, 0.13 mmHg/sec/ml (0.09 - 0.19) vs male, 0.09 (0.06 - 0.15), P=0.039]. There was no sex difference in the mean area of the main PA, suggesting that the main PA was stiffer in female.

3. Lines 380-387: The authors use Zc synonymously with stiffness, but state "lower Zc may represent larger area but not higher stiffness." This contradicts the former statement and needs to be consistent.

Response

We appreciate your suggestion. We revised the manuscript as follows.

<Page 16, Line 376-382>

When compared with normal subjects, wave transmits more quickly in PAH due to ~~higher stiffness of the proximal arteries~~ higher Zc [Su J. et al. 2017]. Surprisingly, however, when wave speed was estimated using Zc, the area of the main proximal PA and blood viscosity (see Eq. 7), mid-systolic returners had slower wave transmission as a consequence of lower Zc (Eq. 4), than late-systolic or diastolic returners. Here, lower Zc may represent larger area but not less stiffness of the proximal arteries

4. Lines 392-394: Wave speed would increase if PAH caused further vascular stiffening. Wave speed would contribute, then, to early reflected wave returns in addition to reflection sites. Both wave speed and reflection sites are important. Please clarify this.

Thank you for your comments We agree thoroughly that both wave speed and reflection sites are important to induce early return of reflected waves. As Su et al. reported, wave speed is accelerated in patients with PAH compared to normal subjects, suggesting that the initial development of PAH may involve the acceleration of wave speed. On the other hand, when analyzing the patients with already developed PAH in the present study, wave speed is not likely to contribute to early return of reflected waves. Based on the previous experimental report, we suggested that once PAH developed, the reflection sites may have the main role to induce the early return of reflected waves. To clarify this point, we revised the manuscript as follows.

<Page 17, Line 387-392>

The initial development of PAH may involve the acceleration of wave speed (Su et al. 2017). With the progress of PAH, wave reflection may occur at closer proximity in pulmonary arteries [Toba M. et al. 2014]. Once PAH is developed, therefore, the reflection sites rather than wave speed may be more important to cause early return of reflected wave. Further clinical investigations are necessary to confirm this explanation.

Dear Dr Westerhof,

Re: JP-RP-2022-282422R2 "When right ventricular pressure meets volume: the impact of arrival time of reflected waves on right ventricle load in pulmonary arterial hypertension" by Masafumi Fukumitsu, Joanne A Groeneveldt, Natalia J. Braams, Ahmed A Bayoumy, J. Tim Marcus, Lilian J. Meijboom, Frances S de Man, Harm-Jan Bogaard, Anton Vonk Noordegraaf, and Berend E. Westerhof

I am pleased to tell you that your paper has been accepted for publication in The Journal of Physiology.

NEW POLICY: In order to improve the transparency of its peer review process The Journal of Physiology publishes online as supporting information the peer review history of all articles accepted for publication. Readers will have access to decision letters, including all Editors' comments and referee reports, for each version of the manuscript and any author responses to peer review comments. Referees can decide whether or not they wish to be named on the peer review history document.

The last Word version of the paper submitted will be used by the Production Editors to prepare your proof. When this is ready you will receive an email containing a link to Wiley's Online Proofing System. The proof should be checked and corrected as quickly as possible.

Authors should note that it is too late at this point to offer corrections prior to proofing. The accepted version will be published online, ahead of the copy edited and typeset version being made available. Major corrections at proof stage, such as changes to figures, will be referred to the Reviewing Editor for approval before they can be incorporated. Only minor changes, such as to style and consistency, should be made a proof stage. Changes that need to be made after proof stage will usually require a formal correction notice.

All queries at proof stage should be sent to TJP@wiley.com

Are you on Twitter? Once your paper is online, why not share your achievement with your followers. Please tag The Journal (@jphysiol) in any tweets and we will share your accepted paper with our 23,000+ followers!

Yours sincerely,

Bjorn Knollmann
Senior Editor
The Journal of Physiology

P.S. - You can help your research get the attention it deserves! Check out Wiley's free Promotion Guide for best-practice recommendations for promoting your work at www.wileyauthors.com/eeo/guide. And learn more about Wiley Editing Services which offers professional video, design, and writing services to create shareable video abstracts, infographics, conference posters, lay summaries, and research news stories for your research at www.wileyauthors.com/eeo/promotion.

*** IMPORTANT NOTICE ABOUT OPEN ACCESS ***

Information about Open Access policies can be found here <https://physoc.onlinelibrary.wiley.com/hub/access-policies>

To assist authors whose funding agencies mandate public access to published research findings sooner than 12 months after publication The Journal of Physiology allows authors to pay an open access (OA) fee to have their papers made freely available immediately on publication.

You will receive an email from Wiley with details on how to register or log-in to Wiley Authors Services where you will be able to place an OnlineOpen order.

You can check if your funder or institution has a Wiley Open Access Account here <https://authorservices.wiley.com/author-resources/Journal-Authors/licensing-and-open-access/open-access/author-compliance-tool.html>

Your article will be made Open Access upon publication, or as soon as payment is received.

If you wish to put your paper on an OA website such as PMC or UKPMC or your institutional repository within 12 months of publication you must pay the open access fee, which covers the cost of publication.

OnlineOpen articles are deposited in PubMed Central (PMC) and PMC mirror sites. Authors of OnlineOpen articles are permitted to post the final, published PDF of their article on a website, institutional repository, or other free public server, immediately on publication.

Note to NIH-funded authors: The Journal of Physiology is published on PMC 12 months after publication, NIH-funded authors DO NOT NEED to pay to publish and DO NOT NEED to post their accepted papers on PMC.

EDITOR COMMENTS

Reviewing Editor:

The authors have responded to all remaining issues that were raised.

Senior Editor:

I concur with the reviewing editor. Excellent work!

REFEREE COMMENTS

Referee #1:

This revision has addressed all previous concerns. This is a well done and interesting contribution to the field.

2nd Confidential Review

15-Mar-2022